Corrected: Publisher correction

# DNA double-strand breaks in telophase lead to coalescence between segregated sister chromatid loci

Jessel Ayra-Plasencia [1,2] & Félix Machín [1,3]

DNA double strand breaks (DSBs) pose a high risk for genome integrity. Cells repair DSBs through homologous recombination (HR) when a sister chromatid is available. HR is upregulated by the cycling dependent kinase (CDK) despite the paradox of telophase, where CDK is high but a sister chromatid is not nearby. Here we study in the budding yeast the response to DSBs in telophase, and find they activate the DNA damage checkpoint (DDC), leading to a telophase-to-$G_1$ delay. Outstandingly, we observe a partial reversion of sister chromatid segregation, which includes approximation of segregated material, de novo formation of anaphase bridges, and coalescence between sister loci. We finally show that DSBs promote a massive change in the dynamics of telophase microtubules (MTs), together with dephosphorylation and relocalization of kinesin-5 Cin8. We propose that chromosome segregation is not irreversible and that DSB repair using the sister chromatid is possible in telophase.

[1] Unidad de Investigación, Hospital Universitario Nuestra Señora de Candelaria, Santa Cruz de Tenerife, Spain. [2] Escuela de Doctorado y Estudios de Posgrado, Universidad de La Laguna, Santa Cruz de Tenerife, Spain. [3] Instituto de Tecnologías Biomédicas, Universidad de La Laguna, Santa Cruz de Tenerife, Spain. Correspondence and requests for materials should be addressed to F.M. (email: fmachin@funcanis.es)

DNA double-strand breaks (DSBs) represent one of the most toxic forms of DNA damage, which, if left unrepaired, leads to cell death. Cells repair DSBs through two major mechanisms: non-homologous end joining (NHEJ) and homologous recombination (HR). Whereas DSB repair prevents cells from dying, inaccurate repair might be a major source of mutagenesis and genomic instability. The budding yeast *Saccharomyces cerevisiae* has served for several decades as one of the most useful model organisms to study both repair mechanisms, including their influence in the stability of the genome. Thus, NHEJ is generally considered error-prone as it often creates short deletions or insertions at the site of the DNA junction[1,2]. In addition, NHEJ can lead to chromosome translocations when two or more DSBs coincide in space and time. By contrast, HR is generally considered an error-free repair mechanism when the intact sister chromatid serves as a template. Nevertheless, the risk of choosing alternative partially homologous sequences during HR repair may actually feed chromosome rearrangements. For instance, the use in diploid cells of the homologous chromosome, instead of the sister chromatid, may result in loss of heterozygosity. Hence, it is not surprising that yeast, and many other organisms, prefers HR only when a sister chromatid is available in close proximity. Cells lack sister chromatids in $G_1$, the resting period of the cell cycle between the segregation of the sister chromatids to the daughter cells and the next replication of the chromosomal DNA. Because $G_1$ is the only cell cycle stage where the activity of the cyclin dependent kinase (CDK) is low, it appears logical that cells have coupled the CDK activity to the selection between NHEJ and HR[3–8]. Accordingly, low CDK activity inhibits HR in favour of NHEJ, whereas high CDK promotes HR. However, there is a small window in the cell cycle, where CDK is high, despite a sister chromatid is not physically available for HR: late anaphase/telophase.

Herein, we address this paradox by studying the cell response to DSBs in telophase. We find that such response resembles in many ways what is seen in $S/G_2$, including the activation of the DNA damage checkpoint (DDC), which leads to a delay in the telophase-$G_1$ transition in this case. Surprisingly, we observe that the segregation of sister chromatids is partly reverted and that sister loci can coalesce after generation of DSBs. We further show that this regression phenotype mechanistically depends on the DDC, as well as the kinesin-5 microtubule motor protein Cin8. We conclude that chromosome segregation can be a reversible process.

## Results

### DSBs in telophase activate the DDC to block the entry in $G_1$.
We took advantage that *S. cerevisiae* cells can be easily and stably arrested in telophase to check the DSB response at this cell cycle stage. We arrested cells in telophase through the broadly used thermosensitive allele *cdc15-2*. Cdc15 is a key kinase in the Mitotic Exit Network (MEN) that allows cytokinesis and reduction of CDK activity, a hallmark of $G_1$[9]. We created randomly distributed DSBs using phleomycin, a radiomimetic drug[10]. Treatment with phleomycin (10 µg mL⁻¹, 1 h) caused hyperphosphorylation of Rad53 (Fig. 1a), a classical marker for the activation of the DDC[11]. The degree of hyperphosphorylation was equivalent to those seen in $G_1$- and $G_2/M$-blocked cells, where Rad53 amplifies the corresponding checkpoint responses that delay $G_1$-to-S transition and anaphase onset, respectively[12]. We thus checked whether a telophase-to-$G_1$ delay was also observed after phleomycin treatment. Indeed, concomitant removal of phleomycin and re-activation of Cdc15-2 (shift the temperature from 37 to 25 °C) showed a delay in both cytokinesis and $G_1$ entry relative to cells which were not treated with phleomycin (mock treatment). For instance, plasma membrane

ingression and resolution at the bud neck (abscission) was clear for ~70% of cells just 1 h after Cdc15 re-activation (Fig. 1b); note that most mother–daughter doublets remain together during a *cdc15-2* release, at least for the upcoming cell cycle[13,14]. When telophase cells were treated with phleomycin, abscission was severely delayed; <50% by 3 h (Fig. 1b). The telophase-to-$G_1$ delay was also evident through three additional markers. Firstly, only $G_1$ cells respond to the alpha-factor pheromone (αF) acquiring a shmoo-like morphology. We thus added αF after reactivating Cdc15 and found that <50% of phleomycin-treated cells had responded by 3 h, versus ~75% of mock-treated cells (Supplementary Fig. 1). Secondly, the CDK-inhibitor Sic1 is only present in $G_1$ cells[15]. We checked Sic1 levels after Cdc15 re-activation and found a clear delay in its production after phleomycin treatment (Fig. 1c). Thirdly, flow cytometry (FACS) showed that the 2C content, expected in cells arrested in telophase, was long-lasting after phleomycin. By contrast, a mock-treated culture shortly turned this 2C peak into either 1C content (a subset of mother–daughter doublets are separated during the harsh treatment for FACS, provided that cytokinesis is completed) or 4C (DNA replication of the immediate progeny without mother–daughter separation) (Fig. 1d). Finally, we confirmed genetically that the DDC was responsible for this telophase-$G_1$ delay. This checkpoint relies on a biochemical cascade that goes from signalling kinases, such as Mec1 and Tel1, to effector kinases such as Rad53 and Chk1[8,16]. In between, the adaptor kinase Rad9 transduces the checkpoint signal from the signalling to the effector kinases; with *rad9Δ* mutants being incapable of blocking the $G_1$-to-S and $G_2/M$-to-anaphase transitions after DSBs[17–19]. In our scenario, *cdc15-2 rad9Δ* mutants failed to block the telophase-$G_1$ transition after phleomycin treatment (Fig. 1e, f), strongly pointing towards an active role of the DDC in this delay.

### Segregated sister loci can coalesce after DSBs in telophase.
We next focussed on the behaviour of chromosome loci during and after phleomycin treatment in the telophase arrest. We first took advantage of the fact that our *cdc15-2* strains also carried YFP-labelled loci along the chromosome XII right arm (cXIIr; tetO/TetR-YFP system)[13]. We started with cXII centromere (cXII-Cen), as a representative of the centromere cluster according to the Rabl configuration[20]. We noted that, whereas mock-treated telophase cells maintained a constant distance of ~8 µm between segregated sister centromeres, phleomycin caused a shortening of this distance to <6 µm (Fig. 2a and Supplementary Fig. 2a). This approximation between sister centromeres occurred around 1 h after phleomycin addition and was maintained for at least another hour upon phleomycin removal. Furthermore, we noted that the approximation was asymmetric (Fig. 2b and Supplementary Fig. 2b), and up to 25% of phleomycin-treated telophase cells had one sister centromere at the bud neck (versus <5% in mock-treated cells; $p < 0.001$, Fisher's exact test). Sister centromere approximation appeared specific to phleomycin (i.e. DSBs) since other DNA damaging agents expected not to generate DSBs in telophase did not bring about this phenotype. Neither methyl methanesulfonate (MMS) nor hydroxyurea (HU) led to cXII-Cen approximation (Fig. 2c). Note that these two agents should generate DSBs only during ongoing DNA replication (S phase), and just after prolonged incubation or in checkpoint-deficient mutants[21]. Accordingly, both agents minimally increased Rad53 phosphorylation in telophase-blocked cells (Supplementary Fig. 3). Furthermore, sister loci approximation was not restricted to centromeres or the extra-long chromosome XII. When we looked at sister loci located in the middle of the longest arm of chromosome V, a representative medium size chromosome, we also observed approximation (Fig. 2d).

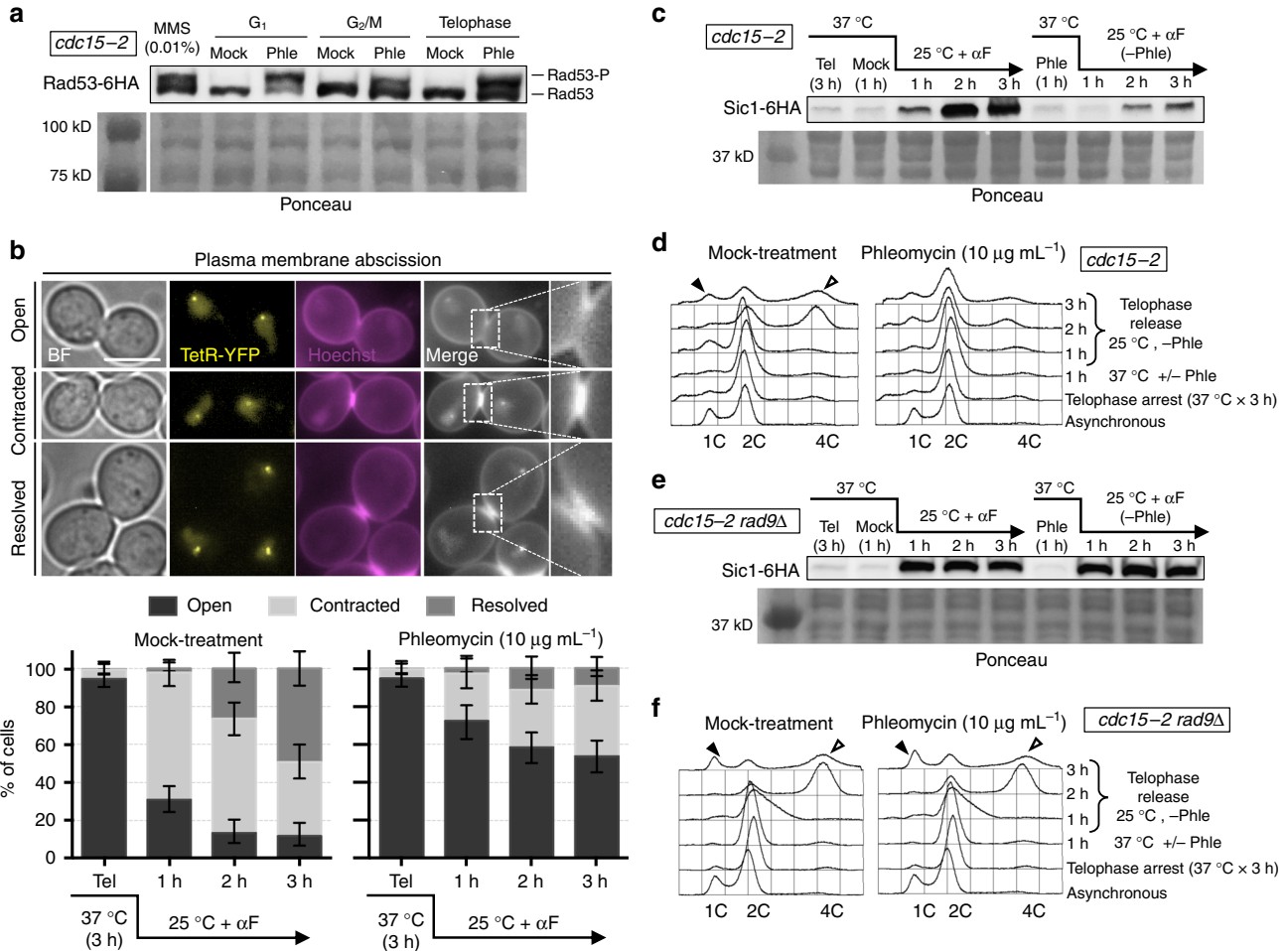

**Fig. 1** Phleomycin triggers the DNA damage checkpoint to delay telophase-G$_1$ transition. **a** Rad53 gets hyperphosphorylated in telophase after phleomycin treatment. Strain FM2329 was arrested at the indicated cell cycle stages. Phleomycin (10 µg mL$^{-1}$) was then added and cells were harvested after 1 h for western blot analysis. MMS (0.01%), added to an asynchronous culture, was used as Rad53 hyperphosphorylation control. The leftmost lane in the Ponceau staining corresponds to the protein weight markers. **b** Cytokinesis is delayed after phleomycin treatment in telophase. Strain FM593 was arrested in telophase, the culture split into two and one subculture treated with phleomycin. After 1 h, phleomycin was washed away and both subcultures were released from the telophase block, taking samples every hour for plasma membrane abscission analysis using Hoechst staining. The G$_1$-blocking pheromone alpha-factor (αF) was added at the time of the telophase release to simplify cell outcomes. On the top, representative micrographs of telophase cells with different degrees of cytokinesis completion. Scale bar corresponds to 5 µm; BF, bright field. At the bottom, charts showing the march of cytokinesis during the telophase release (one representative experiment ± CI95). **c** Sic1 synthesis (G$_1$ marker) is delayed after DSBs in telophase. FM2323 was treated as described in panel (**b**), taking samples for western blot analysis. **d** Cell separation and entry in a new S phase is blocked after DSBs in telophase. FM593 was treated as described in panel (**b**). In this case, though, αF addition was omitted. Samples were taken for FACS analysis. DNA content (1C, 2C or 4C) is indicated under each FACS profile. Filled arrowheads point to the 1C peak; hollow arrowhead points to the 4C peak. **e** Sic1 synthesis is not delayed in strains impaired for the DNA damage checkpoint. FM2477 was treated and samples processed as described in panel (**c**). **f** Cell separation and entry in a new S phase is not blocked after DSBs in telophase in strains impaired for the DNA damage checkpoint. FM916 was treated and samples processed as described in panel (**d**). Source data are provided as a Source Data file

Similar shortening for the distance between sister loci was seen for telomeres (Fig. 3a and Supplementary Fig. 4). Strikingly, we also observed coalescence between sister telomeres (Fig. 3a; ~7% in phleomycin vs ~2% in mock treatment for cXIIr-Tel; $p < 0.001$, Fisher's exact test), which was further confirmed through short-term videomicroscopy (Fig. 3b; Supplementary Movies 1–3). Filming individual cells also showed acceleration of interloci movement and how coalescence lasted longer than expected from simple Brownian motion (Fig. 3b, c). Approximation and eventual coalescence of a fraction of sister loci appeared to be a general phenomenon after DNA damage caused by phleomycin. Firstly, using the histone variant H2A-mCherry, which labels all nuclear DNA,

we confirmed that phleomycin treatment shortens the distance of the bulk of the segregated nuclear masses (Supplementary Fig. 5a). In addition, we observed confined trafficking of segregated DNA across the bud neck (Supplementary Fig. 5b and Supplementary Movies 4–6). This trafficking involved chromatin that appears partly depleted of histones (at least H2A) or is less condensed than the average segregated masses. Strikingly, phleomycin caused the formation of de novo histone-labelled anaphase bridges (Fig. 3d). These bridges included chromatin confined in bulgy nuclear domains (Fig. 3e, Supplementary Fig. 5a and Supplementary Movie 7). We had described before these bulgy bridges in *top2* mutants arrested in telophase and in the *cdc14-1* late anaphase block, but

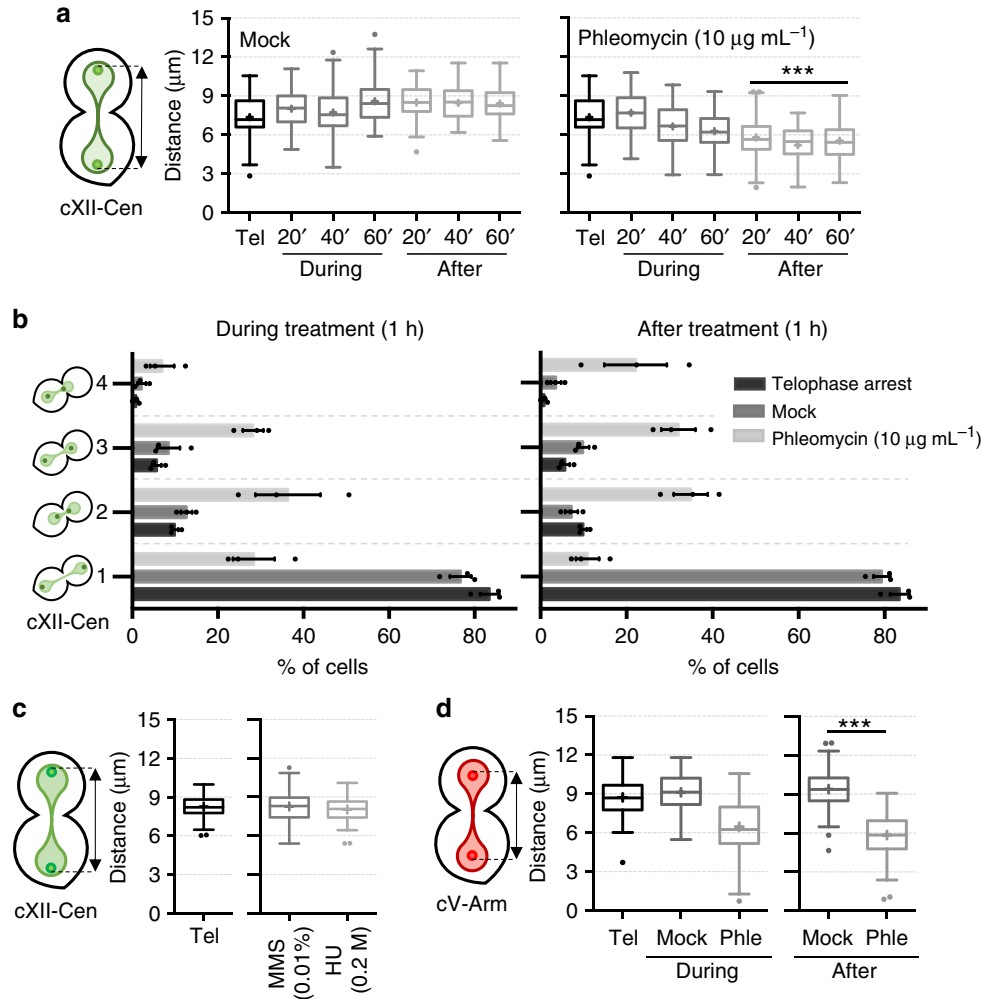

**Fig. 2** DNA double-strand breaks in telophase lead to sister loci approximation. **a** Sister centromeres approach each other during and after generating DSBs in telophase. FM593 was treated as in Fig. 1d, except for the fact that the telophase block was maintained (37 °C) even after phleomycin was washed away. Samples were taken every 20′ during and after phleomycin (or mock) treatment and visualised under the microscope. Distance between sister cXII centromeres was measured and box-plotted for each time point ($N = 70$ cells per box; *** indicates $p < 0.0001$ in mock/phleomycin comparisons at each time point, Mann-Whitney $U$ Test). **b** Relative position of cXII sister centromeres from the former experiment was categorised as depicted on the left (mean ± s.e.m., $n = 3$ independent experiments); category 1, both centromeres at near polar locations; category 2, closer centromeres with symmetrical distances to the bud neck; category 3, closer centromeres with asymmetrical distances to the bud neck; category 4, one sister centromere at the bud neck. **c** DNA damage caused by MMS (0.01 % v/v) or HU (0.2 M) do not trigger approximation of sister cXII centromeres in telophase. FM593 was treated as in (**a**) except for the drug, and the distance between sister cXII centromeres was box-plotted. Only the telophase block and 1 h following drug addition (during treatment) are represented. **d** Sister loci in the middle of chromosome V also get closer during and after phleomycin treatment. FM2456 was treated like in (**a**) and the distance between sister cXII centromeres was box-plotted (*** indicates $p < 0.0001$; Mann-Whitney $U$ Test). Source data are provided as a Source Data file

always arising from metaphase–anaphase transitions[13,22]. Secondly, approximation and coalescence were also observed for the repetitive ribosomal DNA array (rDNA), coated with the rDNA binding protein Net1-eCFP (Fig. 3f and Supplementary Fig. 6).

Having shown that DSBs generated by phleomycin in telophase partially turned back sister chromatid segregation, we next wondered about the specificity of this behaviour. Firstly, we addressed if the sustained telophase block contributed to these outstanding cytological phenotypes. Thus, we added phleomycin to cells normally transiting through telophase. There are at least two critical caveats in this experiment; (i) the fact that phleomycin elicits a $G_2/M$ block in asynchronous cells and (ii) the relatively short duration of telophase. Hence, we performed the experiment in a synchronous $G_1$ release and closely monitored the peak of cells transiting through anaphase; i.e.,

maximizing budded cells with segregating cXII-Cen and a nucleoplasmic bridge, as reported by the soluble pool of TetR-YFP (Supplementary Fig. 7; 120′ from the $G_1$ release). Phleomycin addition at that peak led to a higher proportion of cells with closer cXII-Cen and shorter nucleoplasmic bridges across the bud neck (Supplementary Fig. 7; ~20% vs <5% in the mock treatment; $p < 0.001$, Fisher's exact test). In addition, this experiment also points out that phleomycin blocks the cell cycle in telophase since: (i) fewer cells reached a second cell cycle, as indicated by binucleated dumbbells without a nucleoplasmic bridge (Supplementary Fig. 7a, b)[13] and (ii) almost all Rad53 appeared hyperphosphorylated despite <5% of cells stayed in $G_2/$M (mononucleated dumbbells) 1 h after adding phleomycin (Supplementary Fig. 7c). Secondly, we wondered if approximation and coalescence were specific for DSBs generated via phleomycin. For this purpose, we endonucleolytically cleaved a

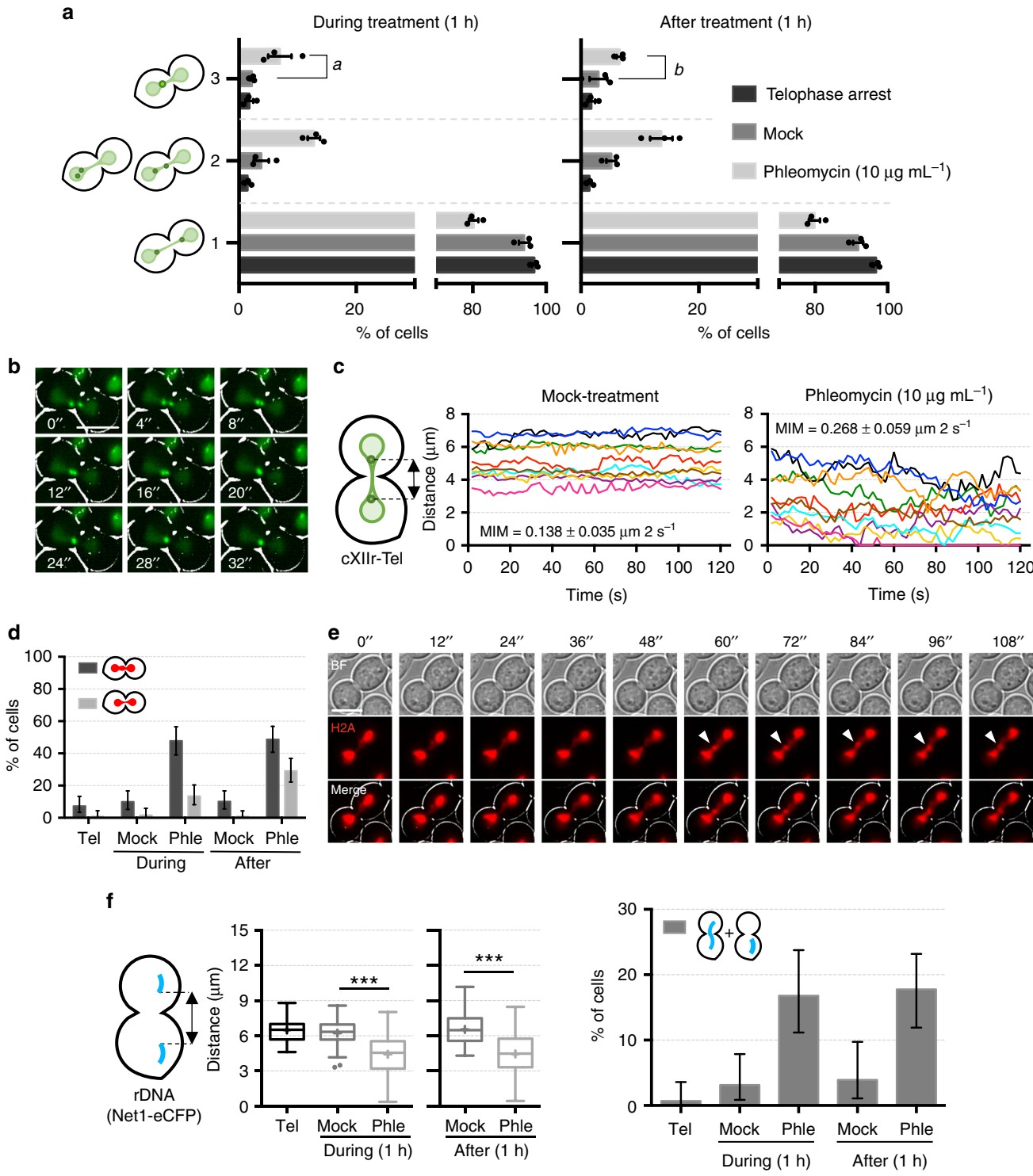

single locus by expressing the I-SceI endonuclease in a strain that carried the I-SceI recognition sequence adjacent to the YEL023C gene, ~44 kbs from chromosome V centromere[23]. This locus has a *tetOs* array on one of its flanks (*URA3* locus), which allowed us to monitor sister loci position before and after I-SceI cleavage. We observed both approximation and coalescence of segregated YEL023C loci after inducing I-SceI for 1 h (Supplementary Fig. 8 and Supplementary Movies 8–10), strongly pointing to coalescence of sister loci as a result of nearby DSBs. Incidentally, we also observed that when one of the loci moved through the bud neck to seek its sister, it stretched to a point where the *tetOs* array

appeared as a line rather than a focus. This striking finding supports the aforementioned statement about local decondensation of chromatin when passing through the bud neck.

Taken together, we conclude in this chapter that all segregated nuclear and nucleolar material gets closer after the generation of DSBs in telophase; and that, under these circumstances, events of sister loci coalescence occur.

**Regression of segregation depends on DDC but not on HR.** Having shown that DSBs in telophase rendered a cell cycle block

**Fig. 3** DSBs in telophase cause sister loci acceleration, coalescence and de novo anaphase bridges. **a** Sister telomeres move to the same cell body and coalesce after phleomycin treatment. FM588 was treated like in Fig. 2a. Relative position of cXIIr sister telomeres was categorised as depicted on the left (mean ± s.e.m., $n = 3$); category 1, sister telomeres in different cell bodies; category 2, telomeres localise in the same cell body or at the bud neck; category 3, sister telomeres coalesce (a [$p < 0.001$] and b [$p = 0.04$] denote the corresponding mock/phleomycin comparisons; Fisher's exact tests from the pooled experiments). **b** Sister telomeres dynamically coalesce during DSBs in telophase. FM588 was treated like in (**a**) and cells were filmed for 2 min after 1 h of phleomycin (or mock) addition. A representative example of coalescence is shown. **c** Inter sister telomere movement accelerates after DSBs. Kinetograms of 10 randomly selected cells. The mean interloci movement (MIM) is displayed within the charts (mean ± s.d., $n = 10$ cells; $p < 0.0001$ in mock/phleomycin comparison, Student's $t$ test). **d** DSBs in telophase generate de novo chromatin bridges. FM2354 was treated like in (**a**) and the histone-labelled nuclear masses categorised in three: binuclear (not shown), with a gross/bulgy bridge (dark grey) and with a thinner/fainter histone-poor bridge (light grey). A representative experiment is shown (±CI95). **e** The de novo chromatin bridges are dynamic. FM2354 was filmed as in (**c**). A representative cell in which a bulgy bridge is dynamically formed from a histone-poor bridge. Filled arrowheads point to the bulge along the chromatin bridge.
**f** Approximation and coalescence also occur for the rDNA/nucleolus. Strain FM2301 was treated like in (**a**). On the left, box-plots of minimum distances between sister rDNA signals under the indicated treatments ($N = 70$ cells per box; *** indicates $p < 0.0001$; Mann-Whitney $U$ Test). Single nucleolar signals were ignored for this calculation. On the right, bar chart for proportion of cells with a single nucleolus either in one cell body or stretched across the bud neck (±CI95). Scale: white bars correspond to 5 μm; BF, bright field. Source data are provided as a Source Data file

at that stage and that segregation is partially regressed to allow sister loci coalescence, we next wondered about the significance of these phenotypes. We began asking whether regression and coalescence were also under control of the DDC. Thus, we checked sister loci approximation, coalescence and acceleration in *rad9Δ* mutants. We found that sister cXIIr-Tel loci did not move to the same cell body and coalesce in this mutant upon phleomycin addition (Fig. 4a). Moreover, interloci acceleration was rather modest in phleomycin (Fig. 4b); whereas in DDC-proficient cells the acceleration was twofold (from 0.14 to 0.27 μm per frame), in *rad9Δ* was only 33% faster (from 0.16 to 0.20 μm per frame). These results demonstrate that the observed cytological responses to DSBs in telophase are regulated by the DDC. Therefore, they are part of the cell reprogramming aimed to cope with DSBs in telophase.

We noticed that dynamic coalescent events between sister loci could represent a chance for HR to repair DSBs with the intact sister chromatid. Therefore, we next drove our attention to HR itself. In yeast, HR is thoroughly impaired by deleting the *RAD52* gene[24]. Thus, we also checked sister loci approximation, coalescence and acceleration in *rad52Δ* mutants. Surprisingly, we found that none of these DSB-induced phenotypes was abolished in *rad52Δ*; rather, cXIIr-Tel coalescence was more frequent than that in the wild type (~20% vs ~7%) (Fig. 4c). Strikingly, not only interloci movement was the highest in *rad52Δ* with phleomycin (0.37 μm per frame) but it was also high even without DNA damage (0.31 μm per frame) (Fig. 4d). The latter suggests that Rad52 plays an unexpected role in restraining loci movement in cells not challenged with exogenously generated DSBs. Lastly, we addressed whether Rad52 may influence the formation of de novo chromatin bridges, finding no differences between the wild type (Fig. 3d) and *rad52Δ* strain (Fig. 4e). Altogether, we concluded that HR itself (Rad52) is not responsible for the aforementioned phenotypes in response to DSBs in telophase. This situates the DDC (Rad9) upstream the observed phenotypes after DSBs, while placing any putative role of HR downstream.

**HR repairs DSBs in telophase.** In order to assess whether HR repairs DSBs in telophase, we performed a series of clonogenic survival experiments. We reasoned that, if DSBs were repaired in telophase using the sister chromatid, sensitivity to phleomycin would be more similar to that of a G₂/M arrest. Conversely, if DSBs are either left unrepaired for the next cell cycle or repaired via NHEJ, the sensitivity would resemble that observed during a G₁ block. Importantly, because HR is chosen for DSBs repair when a sister chromatid is available (i.e. in G₂/M but not in G₁), comparison of survival rates between the wild type and *rad52Δ*

would further inform whether HR is used in telophase for DSB repair. Thus, we arrested both strains in G₁ (αF), G₂/M (Noc) and telophase (*cdc15-2*), and surveyed survival after 1 h of phleomycin treatment (Fig. 4f). We found that (i) survival to DSBs in telophase was similar to G₂/M in the wild type, not G₁; and (ii) Rad52 was directly responsible for such survival since there was a threefold drop of survivors in *rad52Δ* for DSBs generated in both G₂/M and telophase. Taken together, we conclude that DSBs in telophase are repaired by HR during the ensuing arrest and before cells transit into G₁.

**Cin8 drives reversion of chromosome segregation.** Having observed the approximation of segregated sister loci, we next wondered about the cell forces underlying this behaviour. We consequently drove our attention to the spindle apparatus and engineered *cdc15-2* strains where we labelled the microtubules (MTs) (GFP-Tub1) and the spindle pole bodies (SPBs) (Spc42-mCherry), budding yeast equivalent to centrosomes. Phleomycin turned the elongated spindle, characteristic of telophase, into a rather dynamic star-like distribution (Fig. 5a and Supplementary Movies 11–14). This new morphology points to a redistribution of Tubulin towards astral MTs, while nuclear MTs appear misaligned and with a weakened interpolar MT interaction. The change in the spindle morphology shortened the spindle length, which was confirmed by the approximation of the segregated SPBs from ~9 to ~6 μm (Fig. 5b). The separation of SPBs in anaphase pulls attached centromeres apart, favouring the centromere-to-telomere segregation of sister chromatids[25]. Co-visualization of SPBs and cXII sister centromeres showed that SPBs often headed centromeres in the approximation (Fig. 5c), suggesting that either the strengthened astral microtubules or the weakened spindle indirectly drive sister loci approximation by pushing SPBs to each other.

These results led us to check the behaviour of Cin8 upon DSBs in telophase. Cin8 is a bidirectional mitotic kinesin-5 motor protein that makes antiparallel interpolar microtubules slide apart, thus favouring spindle elongation in anaphase[26,27]. Upon phleomycin treatment, Cin8 relocated from the spindle to two discrete foci in telophase-blocked cells (Fig. 6a). These foci likely correspond to SPBs and/or kinetochore clusters[28–30]. Cin8 localization throughout the cell cycle depends on its phosphorylation status, with dephosphorylated Cin8 mostly located at the mitotic spindle[31]. We checked phosphorylation levels of Cin8 after DSBs in telophase and found they are intermediate between S/G₂ (fully dephosphorylated) and an unperturbed telophase (Fig. 6b). Consequently, a partial dephosphorylation of Cin8 occurs upon DSBs in telophase. We also checked if Cdc14, the master phosphatase in anaphase/telophase, played an active role

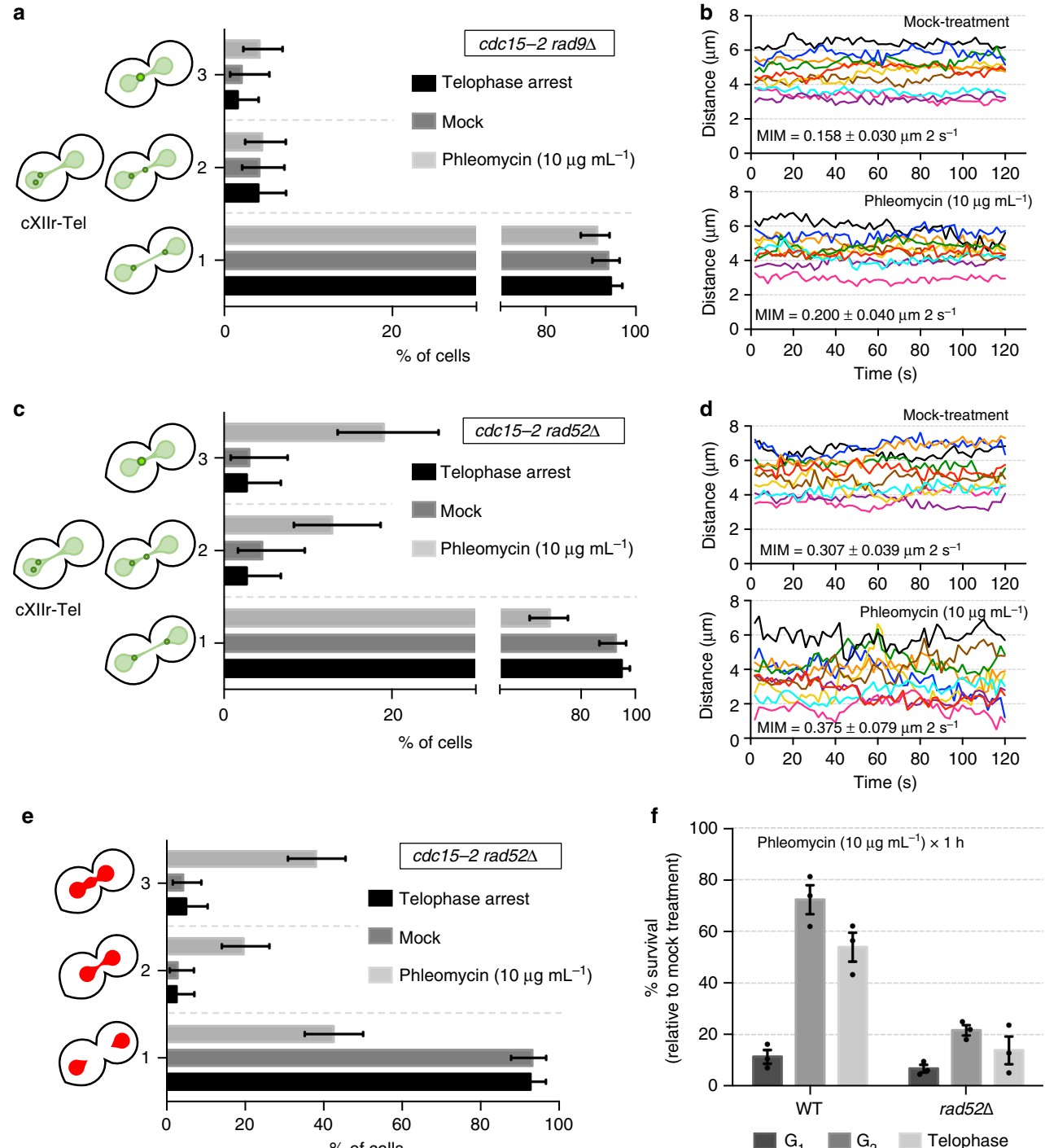

**Fig. 4** Sister loci acceleration and coalescence in telophase depends on Rad9 but is independent of Rad52. **a** Sister telomeres need the DNA damage checkpoint (Rad9 branch) to move to the same cell body and coalesce after DSBs in telophase. FM916 (*rad9Δ*) was treated like in Fig. 2a. Relative position of cXIIr sister telomeres was categorised as depicted on the left (one representative experiment ± CI95); categories 1, 2 and 3 as defined in Fig. 3a. **b** The increase in inter sister telomere movement also depends on Rad9. Kinetograms of 2-min movies from 10 randomly selected cells in the previous experiment. The MIM is also displayed within the charts (mean ± s.d., *n* = 10 cells). Note that the acceleration in interloci movement after phleomycin addition is only one-third of that observed in DDC-proficient cells (compare with Fig. 3c). **c** Sister telomeres coalescence is independent of a functional HR. FM889 (*rad52Δ*) was treated like in Fig. 2a. Relative position of cXIIr sister telomeres was categorised as depicted on the left (one representative experiment ± CI95); categories 1, 2 and 3 as defined in Fig. 3a. **d** Rad52 restrains inter sister telomere movement in telophase. Kinetograms of 2-min movies from 10 randomly selected cells in the previous experiment. The MIM is also displayed within the charts (mean ± s.d., *n* = 10 cells). Note how MIM is already doubled in the mock treatment when compared with the wild type (Fig. 3c). **e** Formation of de novo chromatin bridges is independent of Rad52. Samples from the previous experiments were taken and the nuclear mass stained with Hoechst. Nuclear morphology was categorised as followed: 1, binuclear; 2, thin bridge; 3, gross/bulgy bridge (±CI95). **f** HR repairs DSBs in telophase. Clonogenic survival of strains FM588 (WT) and FM889 (*rad52Δ*) arrested in $G_1$, $G_2$ or telophase before treated with phleomycin (mean ± s.e.m., *n* = 3 independent experiments). Survival is normalised to a parallel mock-treated culture (reference for 100% survival). Source data are provided as a Source Data file

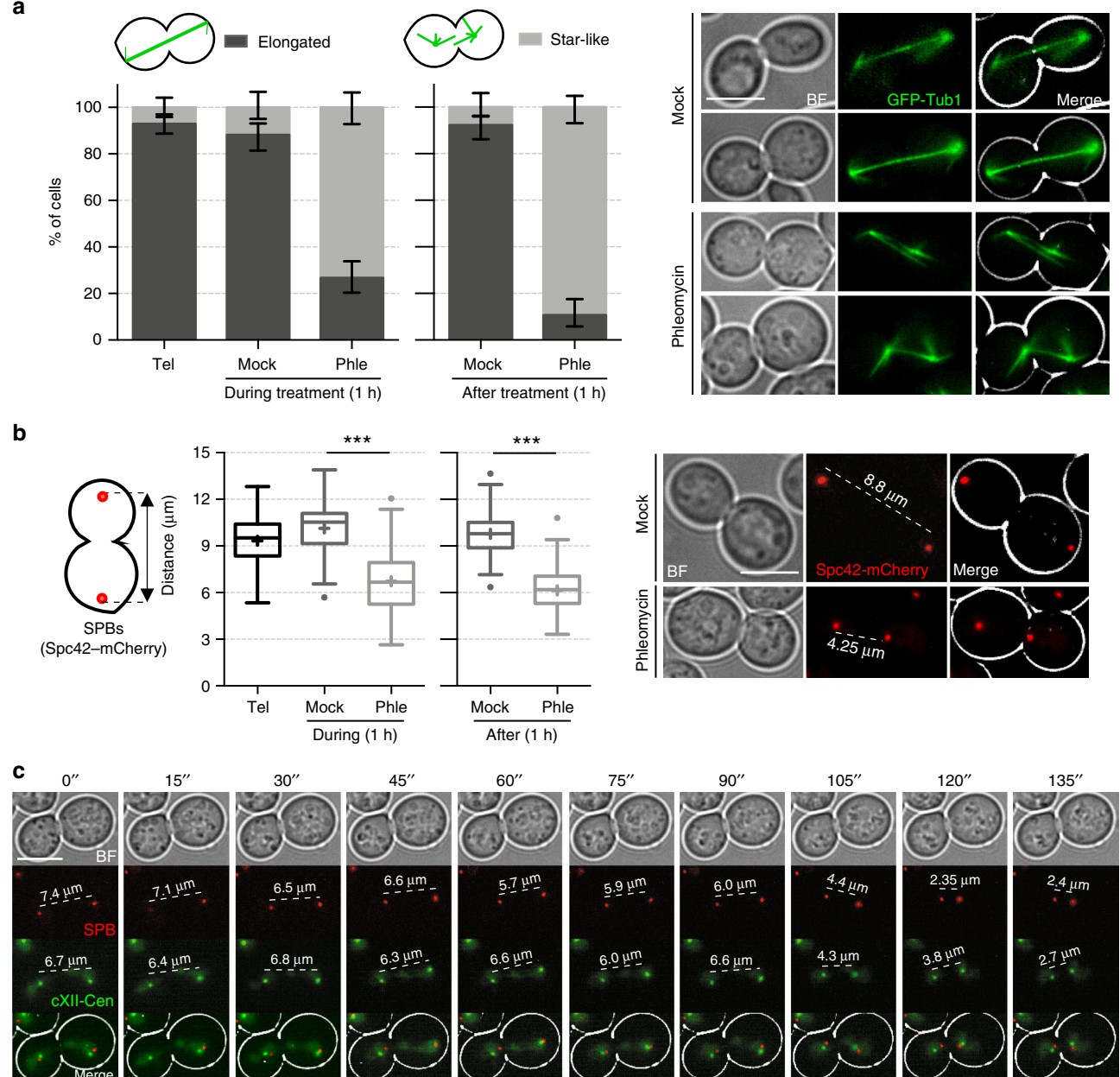

**Fig. 5** Dynamics of microtubules after DNA double-strand breaks in telophase. **a** Microtubules are repositioned after DSBs in telophase. Strain FM2381 was treated as in Fig. 2a. At the indicated conditions, samples were taken and microtubules visualised under the microscope. Two categories were considered for quantification of the spindle: elongated and star-like; the latter being formed by multiple and shorter microtubule fibres arising from each SPB (% cells ± CI95, one representative experiment). **b** SPBs approach each other after DSBs. Strain FM2316 was treated like in Fig. 2a. On the left, box-plots of distances between SPBs under the indicated treatments (*** indicates p < 0.0001; Mann-Whitney U Test). On the right, representative cells for the major phenotypes observed during each treatment. **c** SPBs often move ahead centromeres during the approximation that follows DSB generation in telophase. Strain FM2316 was treated and filmed like in Fig. 3b. Scale: white bars correspond to 5 μm; BF, bright field. Source data are provided as a Source Data file

in dephosphorylating Cin8. Two sequential waves of Cdc14 activation coordinate anaphase events and the telophase-G₁ transition[32]. Cdc14 is activated through its release out of the nucleolus, where it is sequestered for most of the cell cycle. At the *cdc15-2* block, Cdc14 is back in the nucleolus after completion of the first activation wave in early anaphase. It is conceivable, though, that a new partial release upon phleomycin addition could drive Cin8 dephosphorylation. In fact, two previous reports encourage this possibility: Cin8 is a target of Cdc14 in early anaphase[33], and phleomycin promotes Cdc14 release in metaphase-blocked cells[34]. However, we could not observe

Cdc14 release upon phleomycin in *cdc15*-arrested cells (Supplementary Fig. 9a). Furthermore, Cin8 still became dephosphorylated upon phleomycin in the *cdc14-1* mutant (Supplementary Fig. 9b).

We hypothesised that the Cin8 relocalization was a consequence of its novel minus-end-directed motility[27], which might reset the spindle to revert its elongation in cells already in late anaphase. To gain insight on this, we looked at how a set of Cin8 phosphomutants in the motor domain responded to phleomycin during the telophase block. In the Cin8-3A mutant, which mimics a constitutive non-phosphorylated Cin8 in the motor domain (i.e,

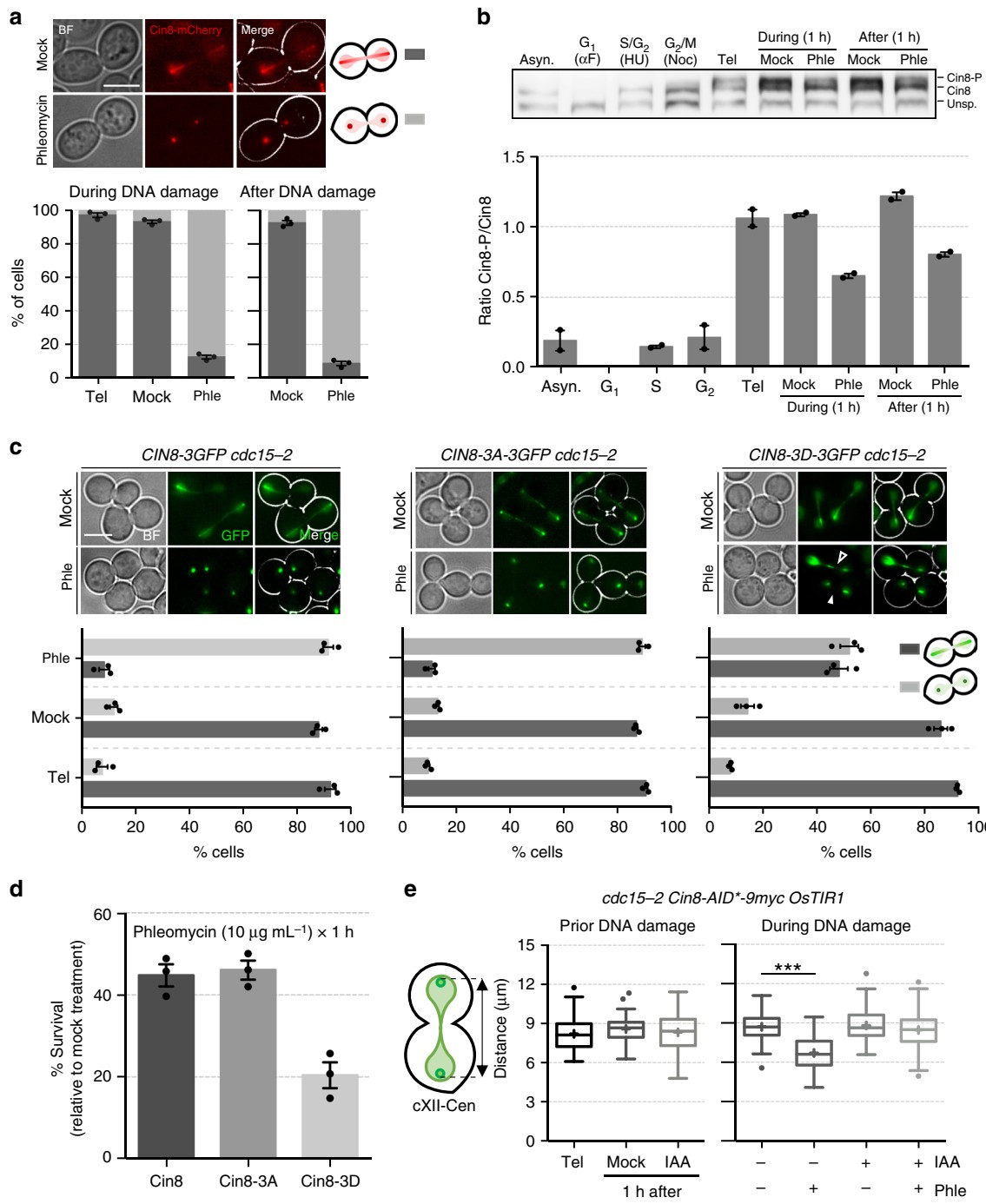

**Fig. 6** Kinesin-5 Cin8 drives sister loci approximation in telophase. **a** Cin8 relocalises from the spindle into SPBs/kinetochores after DSBs. Strain FM2317 was treated like in Fig. 2a and checked under the microscope at indicated times. Location of Cin8 was categorised into two: all along the spindle (dark grey bars) and concentrated in two foci (light grey bars) (mean ± s.e.m., *n* = 3 experiments). **b** A pool of Cin8 becomes dephoshorylated in telophase after DSBs. Strain FM2335 was treated like in Fig. 2a and, in addition, blocked at the indicated cell cycle stages. The upper picture shows western blot of Cin8-9myc, where myc signal appears as a triplet as reported before[53]. The lowest band is unspecific and serves as a loading control. The highest (slowest in PAGE migration) corresponds to the Cin8 phosphorylated forms (Cin8-P). Cin8 is absent in G_1[53]. The lower chart depicts quantification of the Cin8-P/Cin8 ratios (mean ± s.e.m., *n* = 2 independent experiments). **c** The relocalization of Cin8 upon DSBs is partly dependent on its motor domain. Relocalization of WT Cin8 (FM2505) was compared with Cin8 phosphomutants for its motor domain (Cin8-3A, FM2506; Cin8-3D, FM2507). All strains were treated as in Fig. 2a. Upper images show representative micrographs during treatment (1 h). Lower charts depict quantifications (mean ± s.e.m., *n* = 3 independent experiments). **d** Relocalization of Cin8 contributes to cell tolerance to DSBs in telophase. Clonogenic survival of strains in (**c**). Cells were arrested in telophase before being treated with phleomycin (mean ± s.e.m., *n* = 3 independent experiments). Survival is normalised to a parallel mock-treated culture (reference for 100% survival). **e** Cin8 is needed to bring closer sister loci upon double-strand breakage in telophase. FM2466 (bearing a conditional degron Cin8-aid variant) was treated as in Fig. 2a, except for the fact that Cin8-aid degradation was induced 1 h before mock/phleomycin treatments. Scale: white bars correspond to 5 μm; BF, bright field. Source data are provided as a Source Data file

active motility along MTs), phleomycin still led to Cin8 relocalization (Fig. 6c). Nonetheless, only partial relocalization was obtained with the Cin8-3D mutant, which mimics a constitutive phosphorylated Cin8 (i.e., reduced binding to and motility along MTs). The most obvious conclusion from this experiment is that phleomycin polarises Cin8 movement towards the SPBs through its minus-end motility, using MTs as motorways, while it also drives Cin8 dephosphorylation to recruit soluble Cin8 to MTs. This two-step mechanism, recruitment to MTs and walking towards the SPBs, suggests that Cin8 relocalization could play an active role at the poles. In agreement with this, we found that clonogenic survival after phleomycin treatment in telophase was dependent on the Cin8 ability to get dephosphorylated, with Cin8-3D having worse survival than wild-type Cin8 and Cin8-3A (Fig. 6d). In order to confirm the active role hypothesis, we employed a conditional degron version of Cin8 (Cin8-aid) and exposed cells to phleomycin. Indeed, without Cin8, telophase cells were unable to bring closer segregated genetic material (Fig. 6e).

Finally, we addressed whether the shift in MT distribution resulted in an active mechanism for the approximation of the segregated material. We reasoned that Cin8 relocalization weakens interpolar MTs, and perhaps also enforces astral MTs, which, in turn, would favour new pulling forces to bring closer SPB/kinetochores/centromeres. With this aim, we studied the consequences of eliminating MTs in telophase-blocked cells, with or without concomitant DSBs. Nocodazole, a microtubule depolymerizing drug, mimicked most of the phenotypes just described for phleomycin; i.e., shortening of sister loci distances and acceleration of interloci movement (Fig. 7a–c). In general, nocodazole masked the effect of phleomycin. For instance, nocodazole led to a more symmetric approximation of sister loci (compare categories 2 and 3 between Figs. 2b and 7b), with double phleomycin–nocodazole treatment resembling the phenotype of just nocodazole. A similar relationship was seen for those cells where sister centromeres ended up within the same cell body (category 5 in Fig. 7b). This was a very rare event in cells just treated with phleomycin, likely because astral MTs prevented SPBs from passing through the bud neck (Supplementary Movies 13 and 14). Importantly, nocodazole and phleomycin had an additive effect on loci movement for a subset of cells (Fig. 7d), demonstrating that other cell components aside from MTs participate in loci acceleration after the generation of DSBs in telophase. Altogether, these results position weakening of interpolar MTs on top of reinforcing astral MTs as the main cell force that partially regress sister chromatid segregation. Noteworthy, phleomycin did not depolymerise microtubules. Firstly, the effect of nocodazole and phleomycin in telophase MTs was clearly different; nocodazole caused GFP-Tub1 to appear homogenously distributed throughout the cell, with signs of neither nuclear nor astral microtubules (Supplementary Fig. 10a). Secondly, when added to an asynchronous culture, phleomycin arrested cells in $G_2/M$ with the characteristic metaphase spindle (Supplementary Fig. 10b).

## Discussion

In this study, we demonstrate that DSBs can partially regress chromosome segregation in late anaphase. The results shown above question the irreversible nature of chromosome segregation, at least in budding yeast. Importantly, we also provide mechanistic bases for this regression (Fig. 8): (i) weakening of the elongated spindle, likely through dephosphorylation-dependent relocalization of the bipolar kinesin-5 Cin8, which allows sister loci to get closer; (ii) local decondensation of chromatin, which favours passage through the bud neck (i.e., cytokinetic plane); and

(iii) acceleration of loci movement, which increases the probability of closer sister loci to coalesce. Furthermore, we provide evidence that these processes depend on the activation of the DDC to DSBs. We hypothesise that sister loci coalescence in telophase provides a chance to repair DSBs through the efficient and error-free HR pathway; a hypothesis supported by the marked drop of survivors in HR-deficient strains (Fig. 4f). A time window for such repair exists as we also demonstrate that DSBs delay cytokinesis and telophase-$G_1$ transition for more than 2 h (Fig. 1). This telophase checkpoint also depends on Rad9. Even though we have not mechanistically addressed in detail this checkpoint, it is likely that the axis that connects Rad9/Rad53 with MEN inhibition is responsible for the cell cycle arrest[35,36].

DSBs generated in anaphase/telophase have been barely studied despite they pose an even higher risk for genome integrity than those generated in $G_1$, S phase, $G_2$ and prophase/metaphase (M-phase). The reasons for this lay in both, the absence of a nearby sister chromatid and the increased risk of having DSB ends in different compartments (daughters nuclei). In our study, we have generated DSBs once the cells were already in telophase (cdc15-2 block). This scenario physically resembles $G_1$ (no sister chromatid nearby but the two DSB ends locate in the same compartment), yet it shares with $S/G_2/M$ the high CDK activity that favours HR for repair. We generated DSBs through two different approaches. On the one hand, the radiomimetic drug phleomycin, whose major advantage resides in that DSBs are random. This is critical because the probability of having two DSBs in the same pair of sister loci is virtually zero. In this way, we assure that the intact sister chromatid may serve as a genuine template for HR. On the other hand, we also used DSBs generated by the controlled expression of the I-SceI endonuclease. Whereas this approach has the advantage of restricting the DSB to a defined region, which can then be followed by fluorescent tags, it often generates DSBs in both sister loci. Either way, we observed coalescence of selected sister loci (Fig. 3 and Supplementary Fig. 8). In both cases, coalescence was relatively low in end-point experiments (~10% of telophase cells), yet significantly higher than the background levels seen without DSBs (2–3%). This is somehow expected for randomly generated DSBs (phleomycin) since only a minor proportion of cells would have a DSB near the sister loci being monitored by tags. In the case of endonucleolytic cleavage, even though coalescence might appear lower than expected, we must bear in mind that most cells would carry two equivalent DSBs, one per sister chromatid; and this could hinder the capture of an intact sister for HR repair. Because we also filmed individual cells during short periods of sustained DNA damage, we actually suggest that coalescence occurs more often than what we observed in fixed end-point experiments. We reach this conclusion since we could capture dynamic coalescent events in 2-min movies, long (1 h) after the initial DSBs were generated. It is likely that successive DSBs are continuously generated and repaired in telophase under ongoing DNA damage, and transient coalescence reflects cycles of repair attempts of a nearby DSB.

Despite sister loci coalescence was seen at any given time point in ~10% of cells, the approximation of sister loci was almost a general phenomenon after double-strand breakage in telophase (>75% of cells approximate centromeres and SPBs; Figs. 2 and 5). In addition, ~20% of segregated sister loci end up in the same cell body after DSBs, irrespective of whether they then coalesce or not. This implies that one of the labelled sister loci travels back through the narrow bud neck. Accordingly, we observed the appearance of de novo anaphase bridges where before there were binucleated cells (in up to 50% of telophase cells; Figs. 3, 4 and Supplementary Fig. 5). Merging of the segregated nucleoli into a single entity in up to 20% of telophase cells was another indicator of regression in sister chromatid segregation. Altogether, we

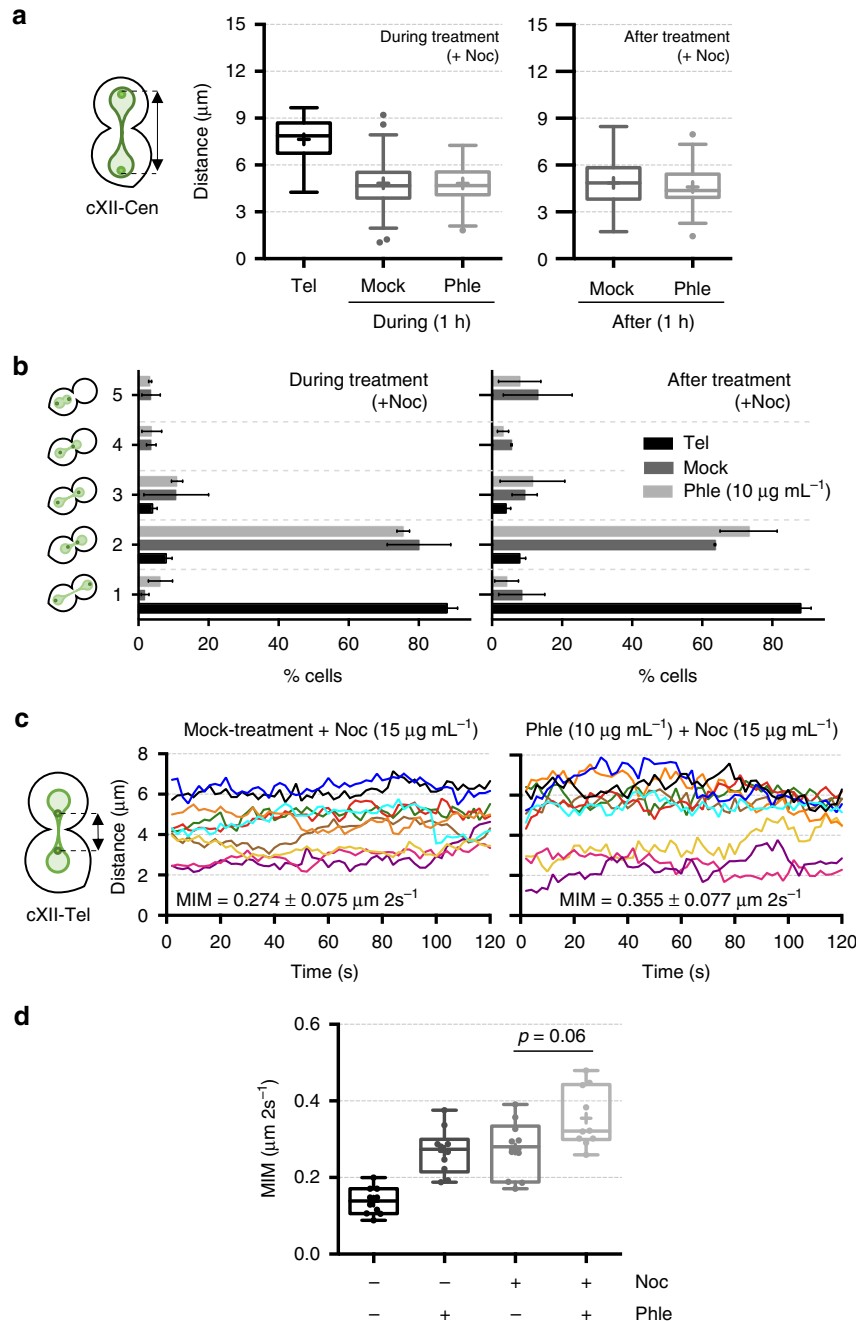

**Fig. 7** Microtubule depolymerization masks the effects of DSBs on sister loci approximation. **a** Sister centromeres also approach each other after depolymerizing telophase microtubules, masking the effect of phleomycin. Strain FM593 was treated as in Fig. 2a, except for the fact that nocodazole (Noc, 15 μg mL$^{-1}$) was added at the time of mock/phleomycin treatments and maintained after that. Distance between sister cXII centromeres was measured and box-plotted as in previous experiments. **b** Relative position of cXII sister centromeres from experiments like in panel (**a**) was categorised as indicated on the left (mean ± s.e.m., $n = 3$ independent experiments). Categories 1–4 are described in Fig. 2b. Category 5, two centromeres laying within the same cell body (very rare in Noc-free experiments). **c** Inter sister telomere movement accelerates upon microtubule depolymerization. Kinetograms of 10 randomly selected FM588 cells previously treated for 1 h with either nocodazole (Noc) alone or Noc plus phleomycin. The MIM during the 2-min movies is also displayed within the charts (mean ± s.d., $n = 10$ cells). **d** Box-plots of MIMs ($N = 10$ cells per box) at the indicated treatment combinations (1 h after addition of the drugs). Note how nocodazole and phleomycin give an additive effect on interloci acceleration for a subset of cells. Source data are provided as a Source Data file

conclude that chromosome segregation in *S. cerevisiae* is more fluid than previously anticipated. This situates cytokinesis, rather than chromosome segregation, as the putative point of no return for using the sister chromatid as template for DSB repair. Whether or not this is extensible to other organisms remains to be determined. Technical caveats greatly difficult such studies since

telophase synchronization is not easily achievable. In metazoans, unlike yeast, there is a clear distinction between $G_2$ and the M-phase. DSBs in $G_2$ lead to immediate cell cycle arrest, whereas DSBs in M-phase lead to distinct responses depending on the type of damage and the model cell line (reviewed in ref. [37]). In all these studies, mitotic DSBs were generated in

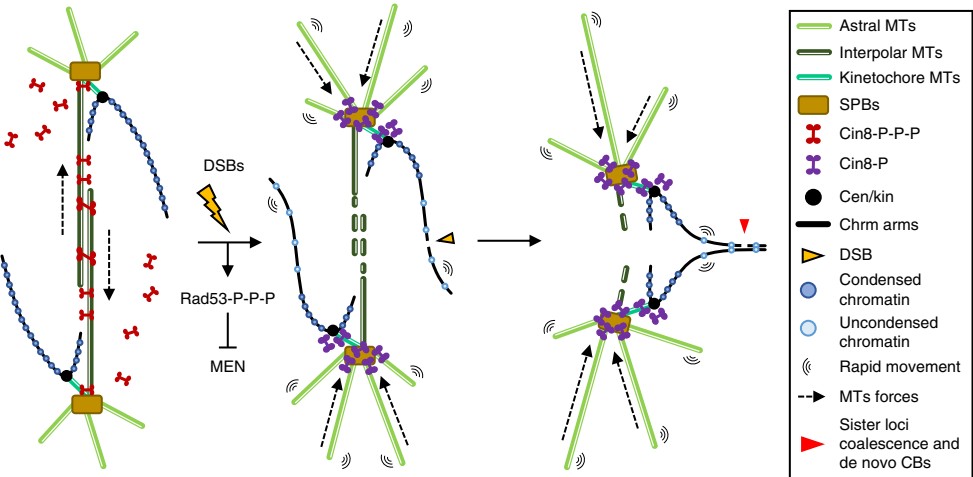

**Fig. 8** Model for the effect of DSBs on segregated sister chromatids. Segregated sister chromatids are maintained away from each other in telophase. Two complementary mechanisms aid to this aim. Firstly, an elongated spindle is maintained by the action of kinesin-5 Cin8 on interpolar microtubules (iMTs). Secondly, segregated sister chromatids are in a hypercondensed state[54]. DSBs locally mobilise the affected chromatin through decondensation, so the histone-poor signal in de novo chromatin bridges, and globally accelerate loci movement. In addition, Cin8 is displaced out of iMTs by partial dephosphorylation, abrogating the spindle forces that keep SPBs far from each other. It is likely that enforced astral MTs also participate in bringing closer the SPBs. All these circumstances make possible for sister loci at chromosome arms to coalesce and repair DSBs through HR with the sister chromatid, even when cells are in telophase

prophase/metaphase. Interestingly, these DSBs in cells committed to mitosis but that have not reached anaphase causes defects in the ensuing chromosome segregation. So far, there has been only one attempt to comprehend how DSBs affect cells transiting through anaphase[38]. In this study, authors used a model marsupial cell line suitable for laser-mediated DNA damage. However, they focused on the consequences for cytokinesis, rather than the behaviour of segregating sister chromatids.

We here also provide mechanistic insights into the regression process. We have found that the spindle undergoes a dramatic change after phleomycin addition (Figs. 5 and 6): (i) the intranuclear spindle collapses and astral MTs appeared reinforced; (ii) the SPBs approximate to each other while keeping themselves in different compartments; and (iii) Kinesin-5 Cin8, a master motor protein for spindle assembly and elongation, fully relocalises to SPBs (and/or kinetochores). We also provide several evidences that all these changes actively contribute to the observed regression and are not simply circumstantial. For instance, sister chromatid approximation requires an active Cin8 localised at the SPBs/kinetochores (Fig. 6). These same players, MTs and Cin8 kinesin-5 orthologs, could be tested in higher eukaryotes if regression were observed after DSBs in anaphase/telophase. Lastly, we demonstrate that Cin8 dephosphorylation in the motor domain is required, yet not sufficient, for full relocalization. It is required because a Cin8 phospho-mimetic version of this kinesin-5 (Cin8-3D) was partially impaired in relocalizing. However, it is not sufficient because a Cin8 phospho-mutant (Cin8-3A) behaved like wild-type Cin8; i.e., (i) it was not enriched at poles in telophase without damage, and (ii) it was fully relocalised to the poles after phleomycin addition. We envision a two-step model to explain these results (Fig. 8). Firstly, Cin8 dephosphorylation recruits soluble Cin8 pools to nuclear MTs, and then, MT-bound Cin8 concentrates at the poles upon further post-translational modifications. These modifications must correspond to residues other than the two serines (S277 and S493) and one threonine (T285) mutated in the Cin8-3D and Cin8-3A variants[31]. Whether the partial dephosphorylation we observed by western blots (Fig. 6) corresponds to dephosphorylation of these three residues or any other(s) that drive Cin8 concentration at the SPBs/kinetochores is presently unknown. Interestingly, the anaphase master phosphatase Cdc14 does not dephosphorylate Cin8 in telophase after DSBs (Supplementary Fig. 9). Cdc14 does dephosphorylate CDK residues in Cin8 at anaphase onset[33]; and S277, T285 and S493 are all CDK residues[31]. Thus, we propose that a phosphatase other than Cdc14 dephosphorylates non-CDK Cin8 residues upon DSBs. A clue for such phosphatase could be found in a recent report whereby a physical interaction between protein phosphatase 1 (PP1) and Cin8 has been described at kinetochores[30].

While Cin8 appears to entirely control sister chromatid approximation, it is likely that coalescence needs more players. We found higher movement of sister loci after DSBs. This kind of movement is oscillatory when comparing the interloci distances, as if pulling and pushing forces were operating successively (Figs. 3, 4 and 7). This increase in oscillatory movement depends on the DDC (Fig. 4). A higher motility of chromatin after DNA damage has been reported before[39,40]. Several nuclear and chromatin rearrangements favour chromatin movement in the confined space of the nucleus, including nucleosomal repositioning, untethering of telomeres from the nuclear envelope and relaxation of the kinetochore-MT interaction[41–44]. It is likely that these processes also contribute to coalescence. Regardless, rapid oscillatory movements and coalescence were highly dependent on Rad9 (i.e., DDC) but not Rad52 (i.e., synapsis between DSB ends and the intact sister chromatid). This raises the possibility that coalescence occurs through a Rad52-independent mechanism, which, nonetheless, is a prerequisite for later execution of HR (Fig. 4f). At present, we can only speculate about the nature of this coalescence facilitator. For instance, very recent findings report that sister chromatid replication is not completed until late anaphase[45]. The maintenance of physical linkages between segregated sister chromatids may catalyse coalescence by zipping sisters from these linkages (e.g., persistent replication forks).

## Methods

**Yeast strains and experimental procedures**. All yeast strains used in this work are listed in Supplementary Table 1. Strain construction was undertaken through standard transformation and crossing methods[46]. C-terminal tags and gene deletions were engineered using PCR methods[46,47]. Strains were grown overnight in air orbital incubators at 25 °C in YEPD media (10 g L$^{-1}$ yeast extract, 20 g L$^{-1}$ peptone and 20 g L$^{-1}$ glucose). To arrest cells in telophase, log-phase asynchronous cultures

were adjusted to OD$_{600}$ = 0.3–0.4 and the temperature shifted to 37 °C for 3 h. In most experiments, the arrested culture was split into two and one of them was treated with phleomycin (10 μg mL$^{-1}$) while the second was just treated with the vehicle (mock treatment). After 1 h incubation, both cultures were washed twice with fresh YEPD and further incubated for 1–3 h to recover from DNA damage. In experiments aimed to check the telophase-G$_1$ transition, temperature was shifted back to 25 °C to allow Cdc15-2 re-activation. To simplify morphological outcomes during the cdc15-2 release, the alpha-factor pheromone (αF) was added after the washing steps unless stated otherwise (50 ng mL$^{-1}$; all strains are bar1, so hypersensitive to αF). For experiments other than telophase-G$_1$ time courses, telophase arrest was maintained after phleomycin/mock treatments by keeping the temperature at 37 °C. Particular experiments such as plasma membrane ingression and responsiveness to αF have been described before[13,22]. Hoechst 33258, utilised to stain both nuclear DNA and plasma membrane, was used at 5 μg mL$^{-1}$. To arrest cells in G$_1$, 50 ng mL$^{-1}$ αF were directly added to an asynchronous culture growing at 25 °C and incubated at that temperature for 3 h. To arrest cells in G$_2$/M, 15 μg mL$^{-1}$ Nocodazole (Noc) was added instead of αF. To arrest cells in S/G$_2$, either 0.01% v/v methyl methanesulfonate (MMS) or 0.2 M hydroxyurea (HU) were added instead. The same concentrations of αF, Noc, HU and MMS were used in experiments where cells were already in telophase before the drug treatment. In experiments with conditional Cin8 degron variants for the auxin system (aid tags)[48], the protein was targeted for degradation by adding 5 mM 3-indol-acetic acid (IAA) 1 h prior to adding phleomycin. Because Cin8 is not essential, we tested the effective degradation range by combining Cin8-aid with kip1Δ (Supplementary Fig. 11); cin8Δ is synthetic lethal with kip1Δ[49,50]. For clonogenic survival assays, log-phase asynchronous cultures were adjusted to OD$_{600}$ = 0.4 before the corresponding arrest and ensuing treatment. After that, 100 μL of 1:10,000 dilutions were spread onto YPD plates. The mock treatments yielded 300–500 CFU/plate in these experiments. Spot sensitivity assays were performed as described before[51]. Briefly, cultures were grown exponentially and adjusted to an OD$_{600}$ = 0.5 and then 10-fold serially diluted in YEPD. A 48-pin replica plater (Sigma-Aldrich, R2383) was used to spot ~5 μL onto the corresponding plates, which were incubated at 25 °C for 3–4 days before taking the photo.

**Microscopy**. A fully motorised Leica DMI6000B wide-field fluorescence microscope was used in all experiments. In time courses, a stack of 20 z-focal plane images (0.3 μm depth) were collected using a 63×/1.30 immersion objective and an ultrasensitive DFC 350 digital camera. Micrographs were taken from freshly collected cells without further processing; 200–300 cells were quantified per experimental data point. Videomicroscopy was also performed in freshly collected cells in a single focal plane (no more than 2 min, time frames of 2 s). The AF6000 (Leica) and Fiji (NIH) softwares were used for image processing and quantifications. The distances between sister loci and SPBs, as well as minimum distances between segregated rDNA and histone-labelled nuclear masses, were measured manually with the AF6000 software. Mean interloci movement (MIM) was calculated from the cumulative absolute variation of distances during videomicroscopy recording divided by the number of frames: $(\Sigma|d_f - d_{f-1}|)/n$; where $d$ is distance, $f$ is the frame number, n is total number of frames, and the summation goes from $f = 2$ to $f = n$. Coalescent events were not considered for calculations.

**Western blots**. For western blotting, 10 mL of the yeast liquid culture were collected to extract total protein using the trichloroacetic acid (TCA) method. Briefly, cell pellets were fixed in 2 mL of 20% TCA. After centrifugation (2500 × $g$ for 3 min), cells were resuspended in fresh 100 μL 20% TCA and ~200 mg of glass beads were added. After 3 min of breakage by vortex, extra 200 μL 5% TCA were added to the tubes and ~300 μL of the mix were collected in new 1.5 mL tubes. Samples were then centrifuged (2500 × $g$ for 5 min) and pellets were resuspended in 100 μL of PAGE Laemmli Sample Buffer (Bio-Rad, 1610747) mixed with 50 μL TE 1X pH 8.0. Finally, tubes were boiled for 3 min at 95 °C and pelleted again. Total proteins were quantified with a Qubit 4 Fluorometer (Thermo Fisher Scientific, Q33227). Proteins were resolved in 10% (7.5% for Rad53 hyperphosphorylation assay) SDS-PAGE gels and transferred to PVFD membranes (Pall Corporation, PVM020C-099). The HA epitope was recognised with a primary mouse monoclonal anti-HA antibody (Sigma-Aldrich, H9658; 1:5,000); and the myc epitope was recognised with a primary mouse monoclonal anti-myc antibody (Sigma-Aldrich, M4439; 1:5,000). A polyclonal goat anti-mouse conjugated to horseradish peroxidase (Promega, W4021; 1:10,000) was used as secondary antibody. Chemiluminescence method was selected for detection, using the ECL reagent (GE Healthcare, RPN2232) and a Vilber-Lourmat Fusion Solo S documentation chamber. The membrane was finally stained with Ponceau S-solution (PanReac AppliChem, A2935) for a loading reference.

**Flow cytometry**. Flow cytometry was performed to determine DNA content[52]. In brief, 1 mL samples were fixed in 75% ethanol. Cells were resuspended in 250 μL 1× SSC buffer containing 0.01 mg mL$^{-1}$ of RNaseA and incubated overnight at 37 °C. Then, 50 μL of 1× SSC containing 1 mg mL$^{-1}$ of proteinase K was added and incubated at 50 °C for 1 h. Finally, 500 μL of 1× SSC with 3 μg mL$^{-1}$ propidium iodide was added and incubated at room temperature for 1 h. BD FACScalibur machine was used to analyse the samples.

**Data representation and statistics**. Bar charts represent proportions of cells which have been categorised (e.g., relative position of sister loci, plasma membrane ingression, etc). Error bars in these charts generally depict the standard error of the mean (s.e.m.), with the aim of quickly showing the interexperimental variability. At least three experiments, performed in different days, were considered for each figure panel. In case only one representative experiment is shown, error bars represent exact 95% confidence interval (CI95) of the proportion. Continuous data (e.g., interloci distance in μm) were represented in box-plots ($N = 100$ cells per box, unless stated otherwise); the centre line depicts the medians, the cross depicts the mean, box limits indicate the 25th and 75th percentile, whiskers extend to the 5th and 95th, and dots represent outliers. R software (https://www.r-project.org/) was used for statistical tests. Differences between experimental data points with continuous data were estimated through a Mann-Whitney $U$ Test. Differences between experimental data points with categorical data were estimated through a Fisher's exact test. In this case, cells counted in all three independent experiments were pooled (>500 cells per data point) to make the contingency tables. All reported $p$ values are two-tailed.

**Reporting summary**. Further information on research design is available in the Nature Research Reporting Summary linked to this article.

## Data availability

The authors declare that all data supporting the findings of this study are available within the paper, its supplementary information, or from the corresponding author upon request. The source data underlying Figs. 1a–c, e, 2a–d, 3a, c, d, f, 4a–f, 5a, b, 6a–e and 7a–c and Supplementary Figs. 1, 3, 4, 5a, 7a, 7c, 8a, 8b and 9b are provided as a Source Data file.

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

## Acknowledgements

We thank Larisa Gheber and Lorraine Symington for yeast strains, and Ivana Malcova for tagging plasmids. We also thank Emiliano Matos-Perdomo and Jonay García-Luis for technical help. This work was supported by Spanish Ministry of Economy, Industry and Competitiveness (research grants BFU2015-63902-R and BFU2017-83954-R to F.M.). Both grants were co-financed with the European Commission's ERDF structural funds.

## Author contributions

F.M. conceived the original project. J.A.-P. performed all the experimental work and prepared the figures. F.M. and J.A.-P. planned and analysed the experiments. F.M. wrote the paper.

## Additional information

**Competing interests:** The authors declare no competing interests.

