## [Peer Review File · Nature Communications]

Reviewers' comments:

Reviewer #1 (Remarks to the Author):

Restoration of a DNA lesion is a key process to ensure cell viability in response to genotoxic stress. The success in the repair of a DNA break relies on two well-conserved pathways known as non-homologous end joining (NHEJ) and homologous recombination (HR). While NHEJ directly re-joins the two ends of the severed DNA, HR requires the presence of a homologous DNA sequence to copy the lost information and fix the lesion. It has been postulated that by coupling HR with Cdk activity, cells initiate this repair pathway only when a sister chromatid is available. However, this model does not explain how cells accomplish DNA repair in late anaphase or telophase, where the Cdk activity is high but the sister chromatids have already been segregated. To circumvent this problem, the authors show in this article that mitotic nuclear segregation is not an irreversible process. Cells with separated nuclei respond to DNA damage by shortening the spindle length in order to bring the sister chromatids back together, a mechanism that opens a window of opportunity for recombinational DNA repair to occur.

To validate this hypothesis, the authors first demonstrate that telophase-arrested cells treated with phleomycin are competent to activate a proficient DNA damage checkpoint. Importantly, under these conditions cells reduced the inter-nuclei distance measured by different approaches. These experiments clearly suggest that in response to DNA damage, telophase segregated nuclei maintain certain capacity to come closer together. Finally, the authors propose that this mechanism relies on the different localization pattern of the kinesin-5 Cin8 along the damage response. Telophase cells treated with phleomycin relocate Cin8 from the spindle to two distinctive foci corresponding to the SPBs or the kinetochores.

This is an interesting work with several observations that explain the paradigm of how cells deal with a DNA lesion generated in a cell cycle stage that simulates HR but in the absence of a sister chromatid. This new concept is an attractive subject that has not been considered before. Besides, the experiments are of high quality and well presented.

However, I have a few concerns about the results. I list here a few suggestions that might help to improve the quality of the manuscript:

1) Is the delay in cell cycle re-entry observed in phleomycin-treated telophase arrested cells exclusively due to the activation of the DNA damage response? Even though there is a clear delay in telophase-G1 progression, I wonder if this checkpoint entirely depends on the canonical damage response. Is it possible to bypass this arrest by using a *mec1Δ sml1Δ* background?

2) To check for sister loci approximation in telophase-arrested cells treated with phleomycin, authors use chromosome XII to measure the distance between sister operators located at centromere and telomere (figure 2a-d), and the localization of the rDNA by using a nucleolar Net1 marker (figure 2g). Taking into account the special features of this chromosome, including its big size and the presence of the nucleolar region, I wonder if the same result can be reproduced by measuring inter-sister distance in other chromosomes.

3) Does the DNA damage checkpoint stimulate sister chromatid coalescence? Could the author avoid inter-nuclei approximation by interfering with the damage checkpoint? Answering this question will reinforce the idea of a DNA damage-dependent mechanism responsible for enhancing nuclear approximation in response to genotoxic stress.

4) The resolution of the Western blot showing Cin8 dephosphorylation after treating cells with phleomycin is of rather poor quality (figure 3e). Authors should try to increase the separation of Cin8 isoforms to show a more convincing dephosphorylation effect after phleomycin addition. Have the authors tried to use Phos-tag containing gels to resolve phosphorylated bands? I recommend

the use of this or other alternative method to increase the separation of the different phosphorylation states of the protein in the Western shown in this figure.

5) It has been postulated that dephosphorylation of Cin8 by Cdc14 in an unperturbed mitosis drives anaphase spindle elongation by enhancing its binding to the midzone. However, the authors propose here that Cin8 dephosphorylation in response to DNA damage excludes the protein from the midzone to binds SPB/kinetochores. How can the authors reconcile these two controversy observations? Taking into account that Cdc14 is activated in response to DNA damage in several organisms, it is tempting to speculate that Cin8 dephosphorylation in DNA damage-induced telophase-arrested cells depends as well on the activity of this phosphatase. Thus, I suggest determining whether Cdc14 has the ability to modulate Cin8 phosphorylation and re-localization along the DNA damage induction of telophase-arrested cells. Another possible explanation is that the DNA damage response collaborates with the steady-state phosphorylation of Cin8 to modulate its relocation. Thus, inactivation of the DNA damage checkpoint should be enough to enhance a constant Cin8 binding to the spindle. Does a *mec1Δ sml1Δ* mutant affect Cin8 phosphorylation and relocation in response to phleomycin?

6) Even though there are strong evidences supporting the existence of a program that stimulates sister chromatids approximation in telophase-arrested cell responding to DNA damage, is not quite clear whether this process constitutes a physiological mechanism that enhances cell viability in response to genotoxic stress. I think the authors should demonstrate that nuclei coalescence of telophase-arrested cells is an active mechanism that stimulates DNA repair. Does a Cin8 hyper-phosphorylation state affect DNA repair or cell viability in response to DNA damage?

Reviewer #2 (Remarks to the Author):

The manuscript by Ayra-Plasencia and Machin makes the interesting report that DNA damage in telophase cause a partial reversal of sister chromosome segregation and they propose that this may allow provide an opportunity of DSB repair in telophase. The authors use a temperature sensitive allele of CDC15 (*cdc15-2*) to arrest cells in telophase follow by treatment with phleomycin to induce DSBs. By various cytological approaches, the authors show that this treatment leads to reduction in the distance between segregated spindle-pole bodies, centromeres, rDNA and telomeres, which is accompanied by phosphorylation of the Rad53 checkpoint kinase and a delay in the telophase-to-G1 transition upon release from the *cdc15-2* arrest. A similar effect is observed after nocodazole treatment. These effects are also accompanied by a change in the organisation of the mitotic spindle and relocalisation of Cin8 to the SPBs.

Although the study documents an interesting observation with potential high impact for our understanding of genome integrity maintenance, the proposed models and physiological relevance are not tested. In my opinion, the following critical issues should be addressed before publication:

1. The hypothesis is that the observed phenomenon would allow cells to repair DSBs in telophase. Testing this hypothesis is central to strengthening the potential impact of the manuscript. This could be tested by low level expression of nuclease (HO or I-SceI) in cells expressing a fluorescently labelled site-specific cut-site, which would facilitate the visualization of the synapsis of the cut-site with the uncut sister chromatid. Alternatively, the authors could compare the survival of WT and *rad52Δ* cells after treatment with phleomycin or IR during telophase to show that breaks are indeed repaired during the *cdc15-2* arrest.

2. Two models are suggested for bringing the segregated sister chromatids together in telophase after segregation in anaphase:

a. Increased movement of the chromosomes. Such increased movement of damaged chromosomes has been shown to depend on Rad9 (DION et al. 2012), which activates the Rad53 checkpoint as also found in telophase by the authors and shown in the model proposed by the authors. This could be tested by use of a *rad9Δ* mutant.

b. Weakening of the spindle by dephosphorylation of Cin8. This could be tested by the use of phosphorylation defective and mimetic mutants of CIN8 (GOLDSTEIN et al. 2017).

3. An alternative explanation for the observed coalescence of segregated sister chromatids in telophase could be that sister chromatids are still connected by unreplicated regions in the *cdc15-2* mutant as suggested by a recent manuscript deposited in BioRxiv (<https://www.biorxiv.org/content/early/2018/09/05/407957>). Upon destabilisation of the mitotic spindle by the DNA damage checkpoint via Rad53 phosphorylation or by treatment with the spindle poison nocodazole, the tethered sister chromatids could recoil due to the tension in ultrafine or chromatin bridges connecting them. This trivial explanation should also be considered and tested experimentally by the authors.

Minor corrections:

4. It be noted that nocodazole also induces Rad53 phosphorylation (DIANI et al. 2009).
5. Page 4, line 7: The FACS profile does not show a transition of the mock-treated cells from 2C to 1C as stated in the text. Rather there is an accumulation of 4C cells. The reason for this should be clarified.
6. Page 5, line 5: I think "H2A2" should be "H2A".
7. Page 5, line 11: The observation of de novo histone-labelled bridges is supporting the hypothesis of the authors that coalescence of sister chromatids in telophase could facilitate recombinational DNA repair immediately before nuclear division, because it has previously been reported that homologous recombination leads to chromatin bridges (GERMANN et al. 2014). It would be easy to test if the de novo bridges depend on recombination by introducing a *rad52Δ* mutation.
8. Figure 2b: What is the difference between "Telophase arrest" and "mock". Please explain in legend.
9. Does the phenomenon of sister chromatid coalescence extend to other types of DNA damage that activate Rad53 without causing DSBs?
10. The color coding of the microtubules in Figure 4c is difficult to distinguish.

References

- Diani, L., C. Colombelli, B. T. Nachimuthu, R. Donnianni, P. Plevani et al., 2009 *Saccharomyces* CDK1 phosphorylates Rad53 kinase in metaphase, influencing cellular morphogenesis. *J Biol Chem* 284: 32627-32634.
- Dion, V., V. Kalck, C. Horigome, B. D. Towbin and S. M. Gasser, 2012 Increased mobility of double-strand breaks requires Mec1, Rad9 and the homologous recombination machinery. *Nat Cell Biol* 14: 502-509.
- Germann, S. M., V. Schramke, R. T. Pedersen, I. Gallina, N. Eckert-Boulet et al., 2014 TopBP1/Dpb11 binds DNA anaphase bridges to prevent genome instability. *J Cell Biol* 204: 45-59.
- Goldstein, A., N. Siegler, D. Goldman, H. Judah, E. Valk et al., 2017 Three Cdk1 sites in the kinesin-5 Cin8 catalytic domain coordinate motor localization and activity during anaphase. *Cell Mol Life Sci* 74: 3395-3412.

Reviewer #3 (Remarks to the Author):

How cells cope with DNA lesions occurring in anaphase/telophase is an underexplored issue that deserves more attention. To explore this important question, the authors employed a very simple approach. They arrested budding yeast cells in late anaphase using the *cdc15-2* thermosensitive allele and exposed them to phleomycin, a DNA damaging agent. Upon Cdc15 reactivation, they observed that exposed cells often failed to proceed through telophase and into G1. In addition, phleomycin exposure leads to frequent mitotic spindle collapse, coalescence of the two nuclei and

newly formed anaphase bridges. From these new observations, the authors speculate that anaphase is reversible to allow double-strand break repair between sister chromatids.

The concept raised by the manuscript is very interesting. But the data are too preliminary. Key controls are missing to rule out more trivial and, at this stage, perhaps more likely explanations. Further work is needed prior considering publication.

Major issues:

The impact of phleomycin is not restricted to DNA and therefore poorly specific. There are many ways to induce double-strand breaks in a more controlled fashion (e.g. HO, ISCEI, Cas9, EcoRI). At least one should be tested.

The *cdc15-2* arrest at 37°C used to synchronize cells in late anaphase/telophase is not neutral. It may generate outcomes (that is artefacts) that would not occur in cells progressing normally from anaphase onset to G1. For that reason, it is essential to test the consequences of DNA damage during anaphase/telophase in cells synchronized through at least another method (e.g. released from a G1 or G2/M arrest).

Does homologous recombination drive the formation of anaphase bridges? The model suggests it. Testing a *rad51* or *rad52* mutant would address this.

To which extent *cdc15-2* arrested cells exposed to phleomycin lose viability? Is there any evidence that the observed anaphase perturbation promotes recovery?

REPLY TO REVIEWERS.

We thank the reviewers for their careful reading of the MS. We have tried our best to respond to all comments. We have also tried to experimentally address all major concerns and feel that the new data have significantly improved the paper. Detailed responses are given below in bold.

General considerations:

- 1) We have shortened the abstract to 150 words to comply with the Journal's guide to authors.
- 2) We have written new chapters (an introductory paragraph and Discussion) to comply with the Journal's guide to authors. Note that we have moved small parts of the text in the original MS to these new chapters.
- 3) In order to accommodate new experimental data, new figures and figure panels have been prepared. The following table shows a correspondence between the original figure panels and the ones found in the revised version.

Original MS	Revised MS	New data relate to...
Fig 1c	Fig S1	
Fig 1d, e	Fig 1c, d	
---	Fig 1e, f (new)	Rev#1 Q1
---	Fig 2c, d (new)	Rev#2 Q9; Rev#1 Q2
Fig S2	Fig 3a	
Fig 2c-g	Fig 3b-f	
---	Fig 4a-f (new)	Rev#1 Q3, 6; Rev#2 Q1, 2a, 3, 7; Rev#3 Q3, 4
Fig 3a-c	Fig 5a-c	
Fig 3d-e	Fig 6a, b	
---	Fig 6c, d, e (new)	Rev#1 Q6; Rev#2 Q2b
Fig 4a, b	Fig 7a, b	
Fig S5a, b	Fig 7c, d	
Fig 4c	Fig 8	
Fig 1c	Fig S1	
Fig S1	Fig S2	
---	Fig S3 (new)	Rev#2 Q9
	Fig S4 (new)	Rev#1 Q2
Fig S3a, b	Fig S5a, b	
Fig S4	Fig S6	
---	Fig S7a-c (new)	Rev#3 Q2
---	Fig S8a-c (new)	Rev#2 Q1; Rev#3 Q1
---	Fig S9a, b (new)	Rev#1 Q5
Fig S5a, b	Fig 7c, d	
Fig S6	Fig S10	
---	Fig S11 (new)	Related to Fig 6e
---	Movies 8-10 (new)	Rev#2 Q1; Rev#3 Q1
Movies 8-11	Movies 11-14	

Reviewer #1 (Remarks to the Author):

Restoration of a DNA lesion is a key process to ensure cell viability in response to genotoxic stress. The success in the repair of a DNA break relies on two well-conserved pathways known as non-homologous end joining (NHEJ) and homologous recombination (HR). While NHEJ directly re-joins the two ends of the severed DNA, HR requires the presence of a homologous DNA sequence to copy the lost information and fix the lesion. It has been postulated that by coupling HR with Cdk activity, cells initiate this repair pathway only when a sister chromatid is available. However, this model does not explain how cells accomplish DNA repair in late anaphase or telophase, where the Cdk activity is high but the sister chromatids have already been segregated. To circumvent this problem, the authors show in this article that mitotic nuclear segregation is not an irreversible process. Cells with separated nuclei respond to DNA damage by shortening the spindle length in order to bring the sister chromatids back together, a mechanism that opens a window of opportunity for recombinational DNA repair to occur.

To validate this hypothesis, the authors first demonstrate that telophase-arrested cells treated with phleomycin are competent to activate a proficient DNA damage checkpoint. Importantly, under these conditions cells reduced the inter-nuclei distance measured by different approaches. These experiments clearly suggest that in response to DNA damage, telophase segregated nuclei maintain certain capacity to come closer together. Finally, the authors propose that this mechanism relies on the different localization pattern of the kinesin-5 Cin8 along the damage response. Telophase cells treated with phleomycin relocate Cin8 from the spindle to two distinctive foci corresponding to the SPBs or the kinetochores.

This is an interesting work with several observations that explain the paradigm of how cells deal with a DNA lesion generated in a cell cycle stage that simulates HR but in the absence of a sister chromatid. This new concept is an attractive subject that has not been considered before. Besides, the experiments are of high quality and well presented.

However, I have a few concerns about the results. I list here a few suggestions that might help to improve the quality of the manuscript:

1) Is the delay in cell cycle re-entry observed in phleomycin-treated telophase arrested cells exclusively due to the activation of the DNA damage response? Even though there is a clear delay in telophase-G1 progression, I wonder if this checkpoint entirely depends on the canonical damage response. Is it possible to bypass this arrest by using a *mec1Δ sml1Δ* background?

We have now addressed whether the telophase-G1 delay after phleomycin treatment depends on the DNA damage checkpoint. We have deleted *RAD9* rather than working with the *mec1Δ sml1Δ* double mutant for three reasons: (i) it is simpler as it only requires a single knockout; (ii) it is more specific for the DNA damage checkpoint since it just covers this Mec1-dependent checkpoint branch (as opposed to the Mec1- Mrc1-dependent replication checkpoint branch); and (iii) the other reviewers asked for similar checkpoint experiments and specifically asked for *rad9Δ*. From our data with *rad9Δ* derivatives we could strongly confirm that the telophase-G1 delay was mediated by the DNA damage checkpoint. In the new Fig 1e we demonstrate this by Western blot (abundance of the G1 marker Sic1) and in the new Fig 1f by FACS.

2) To check for sister loci approximation in telophase-arrested cells treated with phleomycin, authors use chromosome XII to measure the distance between sisters operators located at centromere and telomere (figure 2a-d), and the localization of the rDNA by using a nucleolar Net1 marker (figure 2g). Taking into account the special features of this chromosome, including

its big size and the presence of the nucleolar region, I wonder if the same result can be reproduced by measuring inter-sister distance in other chromosomes.

We have now included two non-cXII chromosomal loci in these analyses. We have used chromosome V, a representative medium size yeast chromosome, and looked at two loci: in the middle of the longest arm (new Fig 2d) and close to the left telomere (new Fig S4). In both cases, phleomycin led to approximation of sister loci.

3) Does the DNA damage checkpoint stimulate sister chromatid coalescence? Could the author avoid inter-nuclei approximation by interfering with the damage checkpoint? Answering this question will reinforce the idea of a DNA damage-dependent mechanism responsible for enhancing nuclear approximation in response to genotoxic stress.

We have used the same *rad9Δ* strain mentioned above. We indeed observed that coalescence, co-localization in the same cell body and interloci movement were not different from the mock-treatment (new Fig 4a & b). This situates the activation of the DNA damage checkpoint as a necessary step for partial regression of chromosome segregation in telophase (see also reply to reviewer #2's Q2a).

4) The resolution of the Western blot showing Cin8 dephosphorylation after treating cells with phleomycin is of rather poor quality (figure 3e). Authors should try to increase the separation of Cin8 isoforms to show a more convincing dephosphorylation effect after phleomycin addition. Have the authors tried to use Phos-tag containing gels to resolve phosphorylated bands? I recommend the use of this or other alternative method to increase the separation of the different phosphorylation states of the protein in the Western shown in this figure.

We have tried other conditions but we have not been able to improve the separation. Because this strain already carries another protein with the myc epitope (Cdc15) plus the presence of the unspecific band, our attempts to better resolve Cin8-P with Phos-tagTM resulted in even more complex patterns of multiple bands. Respectfully, the reviewer should consider that we employed conditions described before (Avunie-Masala et al. *J Cell Sci.* 2011; 124(Pt 6):873-8), where a clear assignment of what is Cin8 unphosphorylated, what is Cin8-P and what is unspecific were reported. We found the expected bands where they were supposed to be and confirmed the absence of Cin8 in G1, which in turns serves as an extra quality control.

5) It has been postulated that dephosphorylation of Cin8 by Cdc14 in an unperturbed mitosis drives anaphase spindle elongation by enhancing its binding to the midzone. However, the authors propose here that Cin8 dephosphorylation in response to DNA damage excludes the protein from the midzone to bind SPB/kinetochores. How can the authors reconcile these two controversy observations? Taking into account that Cdc14 is activated in response to DNA damage in several organisms, it is tempting to speculate that Cin8 dephosphorylation in DNA damage-induced telophase-arrested cells depends as well on the activity of this phosphatase. Thus, I suggest determining whether Cdc14 has the ability to modulate Cin8 phosphorylation and re-localization along the DNA damage induction of telophase-arrested cells. Another possible explanation is that the DNA damage response collaborates with the steady-state phosphorylation of Cin8 to modulate its relocation. Thus, inactivation of the DNA damage checkpoint should be enough to enhance a constant Cin8 binding to the spindle. Does a *mec1Δ sml1Δ* mutant affect Cin8 phosphorylation and relocation in response to phleomycin?

This is an interesting observation. It is true that Cdc14-mediated global Cin8 dephosphorylation has been observed and linked to early anaphase spindle elongation (Rocuzzo et al. *Nat Cell Biol.* 2015; 17(3):251-61). However, it is also true that Cin8 regulation through phosphorylation seems more complex than originally anticipated. It seems that there is a sort of fine-tuning of Cin8 activity/localization that involves not only the degree of phosphorylation but how the phosphorylation status of the different sites combines with each other (Goldstein et al. *Cell Mol Life Sci.* 2017; 74(18):3395-3412). This was briefly mentioned in the original MS and has been now more deeply studied and discussed.

We have now added new experiments with Cin8 phospho mutants for the motor domain (new Fig 6c). We believe that they are quite informative about the possible solution to the controversy raised by the reviewer (see also reviewer #2's Q2b):

1. Cin8-3A mimics the dephosphorylated state. Its motor domain is constitutively active. Cin8-3A-GFP is bound to the spindle in telophase and responds to phleomycin like the wt Cin8, this is, relocalizing to the SPB/kinetochores.
2. Cin8-3D mimics the phosphorylated state. Its motor domain is inactive and its binding to the spindle is known to be weakened. Cin8-3D responds to phleomycin only partially, with many cells not relocalizing the phospho mutant to SPBs.

Taking together, we propose that Cin8 gets dephosphorylated after phleomycin treatment to be recruited to the spindle (this would explain Cin8-3D behaviour). The Cin8 pool already in the spindle, and the new soluble Cin8 recruited to it, is then mobilized towards the SPB. This two-step mechanism would explain why Cin8-3A is not at the SPB before phle treatment. This would also explain why Cdc14-mediated dephosphorylation is not enough for Cin8 relocalization.

Regarding the other issue, whether or not Cdc14 mediates the Cin8 dephosphorylation observed after phle, we now present new experiments that argue against this hypothesis. Firstly, we did not observe Cdc14 release after phle treatment in telophase (new Fig S9a). Secondly, there was no difference in the Cin8-P downshift between *cdc15-2* and *cdc14-1* strains (new Fig S9b). It has been recently reported that phle can release Cdc14 in metaphase-arrested cells (Villoria et al. *EMBO J.* 2017; 36(1):79-101.). We understand that the reviewer suggests that DNA damage would reactivate Cdc14 and this would drive Cin8 dephosphorylation in a similar manner. This could be affordable during a metaphase block; nevertheless, it could be rather risky at a telophase block: any minor Cdc14 release could drive cells out of telophase into G1. Indeed, our results in Fig 1 showing that phle blocked telophase-G1 transition were already suggesting that Phle might actually enhance Cdc14 inhibition rather than favouring its release.

6) Even though there are strong evidences supporting the existence of a program that stimulates sister chromatids approximation in telophase-arrested cell responding to DNA damage, is not quite clear whether this process constitutes a physiological mechanism that enhances cell viability in response to genotoxic stress. I think the authors should demonstrate that nuclei coalescence of telophase-arrested cells is an active mechanism that stimulates DNA repair. Does a Cin8 hyper-phosphorylation state affect DNA repair or cell viability in response to DNA damage?

We have tried to address these issues three-fold. First, we have performed clonogenic assays after transient phleomycin treatments, comparing strains competent or not for HR (*RAD52* vs *rad52Δ*; see also reply to reviewer #2's Q1 and reviewer #3's Q4). We found

that *rad52Δ* is hypersensitive to damage in telophase, pointing towards an active role of HR in maintaining viability at this stage (new Fig 4f; see main text for further details).

The second approach is based on a collection of mutants for the motor domain phosphorylation sites of Cin8, as suggested by the reviewer. We have just shown above that, in terms of Cin8 relocalization, Cin8-3A behaves like wt Cin8 whereas Cin8-3D does not. We measured clonogenic survival with this set of mutants and found that Cin8-3D is hypersensitive to phle treatment in telophase (new Fig 6d).

The third approach was based on a new *CIN8-AID* conditional allele we have introduced in our strains. We found that yeast cell cannot approximate their segregated masses without Cin8 (new Fig 6e). This strongly suggests that Cin8, relocalized to the SPB/kinetochores after DNA damage, plays an active role in bringing closer the sister chromatids.

Reviewer #2 (Remarks to the Author):

The manuscript by Ayra-Plasencia and Machin makes the interesting report that DNA damage in telophase cause a partial reversal of sister chromosome segregation and they propose that this may provide an opportunity of DSB repair in telophase. The authors use a temperature sensitive allele of *CDC15* (*cdc15-2*) to arrest cells in telophase followed by treatment with phleomycin to induce DSBs. By various cytological approaches, the authors show that this treatment leads to reduction in the distance between segregated spindle-pole bodies, centromeres, rDNA and telomeres, which is accompanied by phosphorylation of the Rad53 checkpoint kinase and a delay in the telophase-to-G1 transition upon release from the *cdc15-2* arrest. A similar effect is observed after nocodazole treatment. These effects are also accompanied by a change in the organisation of the mitotic spindle and relocalisation of Cin8 to the SPBs. Although the study documents an interesting observation with potential high impact for our understanding of genome integrity maintenance, the proposed models and physiological relevance are not tested. In my opinion, the following critical issues should be addressed before publication:

1. The hypothesis is that the observed phenomenon would allow cells to repair DSBs in telophase. Testing this hypothesis is central to strengthening the potential impact of the manuscript. This could be tested by low level expression of nuclease (HO or I-SceI) in cells expressing a fluorescently labelled site-specific cut-site, which would facilitate the visualization of the synapsis of the cut-site with the uncut sister chromatid. Alternatively, the authors could compare the survival of WT and *rad52Δ* cells after treatment with phleomycin or IR during telophase to show that breaks are indeed repaired during the *cdc15-2* arrest.

We have actually undertaken both experiments. Firstly, we have made use of strains that cut a single locus with I-SceI, a locus that can be further followed under the fluorescence microscope since it carries *tetOs* and *lacOs* arrays flanking the I-SceI recognition sequence (Oh et al. Cell Rep. 2018; 25(7):1681-1692.e4)¹. We have observed remarkable coalescence events between initially segregated sister loci just after I-SceI cutting (new movies 8-10 and Fig S8). We believe that these new findings strongly support the main conclusion of this study.

¹ Despite the presence of the *lacOs*/LacI-YFP system in this strain, we found that the locus-specific signal got lost at 34-37 °C, the temperature regime required to inactivate Cdc15-2. Therefore, we only followed sister loci coalescence through the *tetOs*/tetR-mRFP system.

Additionally, we have performed clonogenic assays to measure survival after telophase-specific DNA damage and found that overall survival is highly impaired in *rad52Δ* (Fig 4f; see also reply to reviewer #1's Q6 and reviewer #3's Q4).

2. Two models are suggested for bringing the segregated sister chromatids together in telophase after segregation in anaphase:

a. Increased movement of the chromosomes. Such increased movement of damaged chromosomes has been shown to depend on Rad9 (DION et al. 2012), which activates the Rad53 checkpoint as also found in telophase by the authors and shown in the model proposed by the authors. This could be tested by use of a *rad9Δ* mutant.

We have now checked how Rad9 influences the inter sister loci movement upon phleomycin treatment (see also reply to reviewer #1's Q3). Results as shown in the new Fig 4b. Indeed, we found that interloci motion was diminished in the *rad9Δ* strain.

b. Weakening of the spindle by dephosphorylation of Cin8. This could be tested by the use of phosphorylation defective and mimetic mutants of CIN8 (GOLDSTEIN et al. 2017).

We have now used these mutants (strains kindly sent by Larisa Gheber). Results as shown in the new Fig 6c & d. We conclude that Cin8 dephosphorylation recruits Cin8 to the spindle in order to then transfer all nuclear Cin8 to SPBs (see answer to reviewer #1's Q6 for further details). Moreover, viability after DSBs in telophase was diminished in the Cin8-3D mutant, which is partly impaired in Cin8-MT attachment. This confirms that Cin8 relocalization is an important part of the cell response to cope with DSBs in telophase.

In addition, we have included a conditional Cin8-aid. The latter strongly points towards an active role of Cin8 in the SPBs/kinetochores, since its absence preclude sister loci approximation after DNA damage (new Fig 6e). We deeply discuss all these new results in the new MS version.

3. An alternative explanation for the observed coalescence of segregated sister chromatids in telophase could be that sister chromatids are still connected by unreplicated regions in the *cdc15-2* mutant as suggested by a recent manuscript deposited in BioRxiv (<https://www.biorxiv.org/content/early/2018/09/05/407957>). Upon destabilisation of the mitotic spindle by the DNA damage checkpoint via Rad53 phosphorylation or by treatment with the spindle poison nocodazole, the tethered sister chromatids could recoil due to the tension in ultrafine or chromatin bridges connecting them. This trivial explanation should also be considered and tested experimentally by the authors.

We noticed the referred MS after posting ours in BioRxiv. We agree that the presence of DNA links between sister chromatids (e.g., unreplicated tracks) may help in bringing sister loci together. We now mention the results from this MS in the Discussion section. Although it is a putative explanation for helping coalescence, it does not seem to drive coalescence by itself. Just remember we add phleomycin (or cut the DNA) in cells which are previously arrested in telophase, and with the sister loci segregated. To test whether replication intermediates (RI) drive coalescence after DNA damage is not that trivial. Probably the results we obtained with *rad9Δ* (new Fig 4a) stand valid here: The proposed RI are probably stabilized by Rad9, as it has been reporter for those enriched after replication stress; therefore, in *rad9Δ* there would be less RI and more DSBs (as a result of RI collapse). Noteworthy, these DSBs do not lead to coalescence (new Fig 4a). Nevertheless, the interpretation to this is rather complicated at present because it may

well happen that DSBs are not recognized at all because Rad9 also plays a crucial role in the DDR (new Fig 1e, f; 4a, b). Therefore, we believe that a firm experimental demonstration of a role of persistent RIs as a driving force in coalescence is not trivial and beyond the scope of this study.

As a final note, we have here focused on microtubules and kinesin-5 Cin8 as the molecular mediators in bringing back together the segregated sister chromatids. While we believe we have firmly demonstrated their contribution, we agree there are more players to be found. Not only remaining sister chromatid linkages, as the reviewer suggests, but also tethering to the nuclear envelope and/or the reported local decondensation near the bud neck seem good candidates to explore further.

Minor corrections:

4. It be noted that nocodazole also induces Rad53 phosphorylation (DIANI et al. 2009).

We have looked at Rad53 phosphorylation in Nz in several experiments and found a clear difference before and after Phle treatment. Hyperphosphorylation, at least as visualized in our conditions, is only seen after DNA damage but not at the G2/M arrest with Nz. There are many examples in the literature with similar Rad53 behaviour (e.g., **J Biol Chem. 2004; 279(38):39636-44. // EMBO J. 2017; 36(1):79-101. // J Biol Chem. 2003; 278(33):30421-4.**). It is likely that Nz partly phosphorylates Rad53, yet to a degree not resolved in our Western blot conditions (or not seen with the Rad53-HA epitope).

5. Page 4, line 7: The FACS profile does not show a transition of the mock-treated cells from 2C to 1C as stated in the text. Rather there is an accumulation of 4C cells. The reason for this should be clarified.

It is true that the most obvious change is the accumulation of a 4C population. This population arises from *cdc15-2* quadruplets formed after an effective *cdc15-2* release. This is a common pattern observed after a *cdc15-2* block and release experiment (at least in our background; see **Quevedo et al. PLoS Genet. 2012; 8(2):e1002509**), and likely comes from a decoordination between septation and the second cell cycle, as it has been previously described (e.g., **Spellman et al. Mol Biol Cell. 1998; 9:3273-97.**). The formation of a 4C content in these quadruplets indicates that both daughter nuclei have entered a second replication round. Having said that, a minor proportion of cells became cycling 1C singlets in the mock experiment as suggested by the 1C fluctuation observed in the arrest-1h-2h-3h series. The likely origin of these cell subpopulation was mentioned in that paragraph but we have now explained this better. Of note, this three-peak pattern is now better observed in the new FACS with the *rad9Δ* mutant, which is not blocked in telophase upon phle treatment (new Fig 1f).

6. Page 5, line 5: I think "H2A2" should be "H2A". **Corrected.**

7. Page 5, line 11: The observation of de novo histone-labelled bridges is supporting the hypothesis of the authors that coalescence of sister chromatids in telophase could facilitate recombinational DNA repair immediately before nuclear division, because it has previously been reported that homologous recombination leads to chromatin bridges (GERMANN et al. 2014). It would be easy to test if the de novo bridges depend on recombination by introducing a *rad52Δ* mutation.

We have now performed this experiment. The result is shown in the new Fig 4e. The conclusion is that *de novo* bridges do not depend on Rad52. In addition, we have evaluated the overall contribution of Rad52 to the regression of sister chromatid segregation. We found that Rad52, unlike Rad9, does not have an effect in sister loci approximation, coalescence and movement; rather, it might actually contribute to restrain these phenotypes in normal telophase cells (with no exogenous DNA damage) (new Fig 4c, d). We do not know the reason underlying these phenotypes. It may be that *rad52Δ* cells are quite sensitive at detecting endogenously-generated DSBs. Future studies are required to address this.

8. Figure 2b: What is the difference between "Telophase arrest" and "mock". Please explain in legend.

Telophase arrest means "3h at 37 °C", prior to split the culture in two. Mock treatment is after 1 extra hour at 37 °C (4h x 37 °C in total), just for one of the two subcultures (no phleomycin treatment). We now explain this better in that figure legend and Material and Methods.

9. Does the phenomenon of sister chromatid coalescence extend to other types of DNA damage that activate Rad53 without causing DSBs?

We have now performed this experiment. The results are shown in the new Fig 2c & S3. Neither MMS nor HU brought about sister loci approximation. However, Rad53 hyperphosphorylation was weak in telophase after drug addition. This is somehow not surprising since MMS and HU cause DNA damage mainly during S-phase (i.e., damage is coupled to DNA replication).

10. The color coding of the microtubules in Figure 4c is difficult to distinguish.

We have now changed this.

References

- Diani, L., C. Colombelli, B. T. Nachimuthu, R. Donnianni, P. Plevani et al., 2009 Saccharomyces CDK1 phosphorylates Rad53 kinase in metaphase, influencing cellular morphogenesis. J Biol Chem 284: 32627-32634.
- Dion, V., V. Kalck, C. Horigome, B. D. Towbin and S. M. Gasser, 2012 Increased mobility of double-strand breaks requires Mec1, Rad9 and the homologous recombination machinery. Nat Cell Biol 14: 502-509.
- Germann, S. M., V. Schramke, R. T. Pedersen, I. Gallina, N. Eckert-Boulet et al., 2014 TopBP1/Dpb11 binds DNA anaphase bridges to prevent genome instability. J Cell Biol 204: 45-59.
- Goldstein, A., N. Siegler, D. Goldman, H. Judah, E. Valk et al., 2017 Three Cdk1 sites in the kinesin-5 Cin8 catalytic domain coordinate motor localization and activity during anaphase. Cell Mol Life Sci 74: 3395-3412.

Reviewer #3 (Remarks to the Author):

How cells cope with DNA lesions occurring in anaphase/telophase is an underexplored issue that deserves more attention. To explore this important question, the authors employed a very

simple approach. They arrested budding yeast cells in late anaphase using the *cdc15-2* thermosensitive allele and exposed them to phleomycin, a DNA damaging agent. Upon Cdc15 reactivation, they observed that exposed cells often failed to proceed through telophase and into G1. In addition, phleomycin exposure leads to frequent mitotic spindle collapse, coalescence of the two nuclei and newly formed anaphase bridges. From these new observations, the authors speculate that anaphase is reversible to allow double-strand break repair between sister chromatids.

The concept raised by the manuscript is very interesting. But the data are too preliminary. Key controls are missing to rule out more trivial and, at this stage, perhaps more likely explanations. Further work is needed prior considering publication.

Major issues:

1) The impact of phleomycin is not restricted to DNA and therefore poorly specific. There are many ways to induced double-strand breaks in a more controlled fashion (e.g. HO, ISCEI, Cas9, EcoRI). At least one should be tested.

We have now created DSBs through I-SceI cutting and followed an adjacent loci under the microscope (see also reviewer #2's Q1). As it is shown in several movies (new movies 8-10) and Fig S8, these endonucleolytic DSBs also drove sister loci approximation and coalescence.

2) The *cdc15-2* arrest at 37°C used to synchronize cells in late anaphase/telophase is not neutral. It may generate outcomes (that is artefacts) that would not occur in cells progressing normally from anaphase onset to G1. For that reason, it is essential to test the consequences of DNA damage during anaphase/telophase in cells synchronized through at least another method (e.g. released from a G1 or G2/M arrest).

Ideally, DSBs should be induced in cycling cells which are just reaching telophase. Nevertheless, as the reviewer can imagine, this is technically difficult in single cell analysis and unaffordable for experiments that require a pool of cells (e.g., Western blot). From a G1 release it is extremely difficult to have a perfectly synchronous telophase culture. Telophase lasts only a few minutes per cell cycle. It is more likely that most cells are in either metaphase/early anaphase or G1 at any given time point when telophase peaks in the synchronous population. The situation is not going to be better from a G2/M release as it occurs less synchronously. In any case, we have now performed an experiment in which Phle was added just at anaphase onset in a culture coming from previous G1 block in conditions that kept Cdc15 active throughout (25 °C). We found that phle (i) still elicited in strong DDR response (Rad53 hypersphosphorylation); and (ii) arrested cells in anaphase/telophase with closer sister chromatids more often than in a mock experiment (new Fig S7).

3) Does homologous recombination drive the formation of anaphase bridges? The model suggests it. Testing a *rad51* or *rad52* mutant would address this.

We have experimentally tested this hypothesis (see also reviewers 2's Q7). We found that a functional HR (we compared WT with *rad52Δ*) is not needed for regression of sister chromatids segregation, including *de novo* formation of anaphase bridges (new Fig 4c, d and f). However, Rad9 (i.e., the DNA damage checkpoint) is needed (new Fig 4a, b). This position sister chromatid regression downstream the DNA damage checkpoint but upstream HR. Noteworthy, HR does repair DSBs in telophase as we now demonstrate with

clonogenic assays (new Fig 4f; see below for details).

4) To which extent *cdc15-2* arrested cells exposed to phleomycin lose viability? Is there any evidence that the observed anaphase perturbation promotes recovery?

This question connects to reviewer #1's Q6 and reviewer #2's Q1. We performed several clonogenic assays after transient Phle treatment that strongly establish that the observed regression aids the cell to cope with DSBs in telophase (new Figs 4f and 6d):

1. DSBs in telophase in the WT *cdc15-2* strain caused viability to drop like DSBs generated in G2/M, but not G1 (which yielded the lowest). This points out that DSBs are repaired in telophase and not left unrepaired for the ensuing G1.
2. The viability of the *rad52* Δ derivative dropped to values seen after DSBs in G1, and mimicked the drop observed after DSBs in G2/M. This confirms that HR plays a central role in repairing DSBs in telophase.
3. Cin8-3D, a phospho-mimetic mutant unable to relocalize to SPBs after DSBs (a step needed for regression) had a worse viability when compared to other Cin8 variants able to respond to DSBs.

REVIEWERS' COMMENTS:

Reviewer #1 (Remarks to the Author):

In my previous revision I raised two main concerns about chromosome coalescence in telophase cells responding to DNA damage and its implication in genome integrity: 1) whether this phenomenon was part of the canonical DNA damage response pathway, and 2) determine the importance of nuclear coalescence in the repair of DNA lesions produced in telophase. To answer the first question, the authors have incorporated a wide range of experiments demonstrating that nuclear coalescence in response to DNA damage is a phenomenon that directly depends on the DNA damage checkpoint. Regarding the second question, the authors have demonstrated that cell survival in response to phleomycin depends on chromosome coalescence by a mechanism that relies on the Cin8 kinesin. In this new revised manuscript the authors have thoroughly addressed my major concerns from the initial submission, and also other minor points that I raised during the revision process. Thus, I recommend the actual version of the manuscript for publication.

I have just minor comments on the revised manuscript:

1) Page 8, line 158. When addressing nucleolar coalescence in response to phleomycin, the text points to fig. 3f and 3g. However, 3g is missing in both the figure and figure legend information.

2) In the "Regression of chromosome segregation depends on an active DNA damage..." section, there are a few sentences where the expression "DDR" has been misused. When discussing about the *rad9Δ* mutant, it is better to use the concept "DNA damage checkpoint" than DDR:

Page 10, line 207. ...whereas in DNA damage checkpoint proficient cells...

Page 10, line 210. ...responses to DSBs in telophase are regulated by the DNA damage checkpoint.

Page 11, line 225. ... This situates the DNA damage checkpoint...

Reviewer #2 (Remarks to the Author):

The manuscript by Ayra-Plasencia and Machin has been revised so that my concerns for the previous version of the manuscript have been addressed. The authors have conducted additional experiments to (1) demonstrate the physiological importance of sister chromatid coalescence in telophase, and (2) test the role of DNA damage checkpoint signalling (RAD9) and Cin8 phosphorylation in sister chromatid coalescence thereby strengthening the proposed model. Further, the authors have made appropriate improvements to the text, which has improved the readability and clarity of the manuscript. I have only a few minor suggestions for corrections that should be considered before publication:

1. Having shown in the revised manuscript that MMS and HU do not cause sister chromatid coalescence in telophase, despite at least MMS causing damage to the DNA, I suggest that the title of the manuscript is misleading. "DNA damage in telophase ..." could be changed to "DNA double-strand breaks in telophase ..." to accommodate this concern.

2. Line 51: It is incorrect that interhomologue recombination always results in LOH; it depends on the segregation pattern of the recombinant chromosomes. I therefore to insert a "may" in front of "results".

3. Do the authors know if MMS activates Rad53 phosphorylation in telophase? If not, that could explain the lack of sister chromatid coalescence after MMS treatment.

4. Line 275: I suggest to delete the word "all" since the authors show that Cdc14 is not responsible for Cin8 dephosphorylation.

5. Line 298: The word "must" is an overstatement. The role of Cin8 could also be passive. I

therefore suggest to change "must" to "could".

6. Line 419: I disagree that the authors have shown that Cin8 relocalization is "sufficient" for the observed regression. The authors have not presented conditions where Cin8 relocalization takes place but none of the other events.

7. Line 429-430: What is the evidence for "further post-translational modifications" of Cin8?

Reviewer #3 (Remarks to the Author):

This revised version is overall a much better and a much more convincing paper. My main concerns were appropriately addressed.

I only advise to place the important ISceI result (currently figure S8) within one of the main figures.

REPLY TO REVIEWERS.

We thank the reviewers for this second round of revision of our MS and the overall positive responses to the revised version. We have amended the suggested minor changes. Detailed responses are given below in bold.

Note that there are other minor changes in the MS to comply with editorial policies: (i) A few sentences have been shortened or erased (to fit with a total length of 5,000 words or fewer), (ii) the results subheadings have been shortened to fewer than 60 characters, and (iii) figure legends have been shortened to fewer than 350 words.

Reviewer #1 (Remarks to the Author):

In my previous revision I raised two main concerns about chromosome coalescence in telophase cells responding to DNA damage and its implication in genome integrity: 1) whether this phenomenon was part of the canonical DNA damage response pathway, and 2) determine the importance of nuclear coalescence in the repair of DNA lesions produced in telophase. To answer the first question, the authors have incorporated a wide range of experiments demonstrating that nuclear coalescence in response to DNA damage is a phenomenon that directly depends on the DNA damage checkpoint. Regarding the second question, the authors have demonstrated that cell survival in response to phleomycin depends on chromosome coalescence by a mechanism that relies on the Cin8 kinesin. In this new revised manuscript the authors have thoroughly addressed my major concerns from the initial submission, and also other minor points that I raised during the revision process. Thus, I recommend the actual version of the manuscript for publication.

I have just minor comments on the revised manuscript:

1) Page 8, line 158. When addressing nucleolar coalescence in response to phleomycin, the text points to fig. 3f and 3g. However, 3g is missing in both the figure and figure legend information.

This has been corrected in the body text.

2) In the “Regression of chromosome segregation depends on an active DNA damage...” section, there are a few sentences where the expression “DDR” has been misused. When discussing about the rad9 Δ mutant, it is better to use the concept “DNA damage checkpoint” than DDR:

Page 10, line 207. ...whereas in DNA damage checkpoint proficient cells...

Page 10, line 210. ...responses to DSBs in telophase are regulated by the DNA damage checkpoint.

Page 11, line 225. ... This situates the DNA damage checkpoint...

We have changed this as suggested. We have also double-checked all references to DNA damage checkpoint/response in the rest of the MS and changed them when appropriate.

Reviewer #2 (Remarks to the Author):

The manuscript by Ayra-Plasencia and Machín has been revised so that my concerns for the previous version of the manuscript have been addressed. The authors have conducted additional experiments to (1) demonstrate the physiological importance of sister chromatid coalescence in telophase, and (2) test the role of DNA damage checkpoint signalling (RAD9) and Cin8 phosphorylation in sister chromatid coalescence thereby strengthening the proposed model. Further, the authors have made appropriate improvements to the text, which has improved the readability and clarity of the manuscript. I have only a few minor suggestions for corrections that should be considered before publication:

1. Having shown in the revised manuscript that MMS and HU do not cause sister chromatid coalescence in telophase, despite at least MMS causing damage to the DNA, I suggest that the title of the manuscript is misleading. "DNA damage in telophase ..." could be changed to "DNA double-strand breaks in telophase ..." to accommodate this concern.

We agree and have changed the MS title accordingly. We have also referred to DSBs, rather than DNA damage, in the body text when it seems more accurate to do so.

2. Line 51: It is incorrect that interhomologue recombination always results in LOH; it depends on the segregation pattern of the recombinant chromosomes. I therefore to insert a "may" in front of "results".

Corrected.

3. Do the authors know if MMS activates Rad53 phosphorylation in telophase? If not, that could explain the lack of sister chromatid coalescence after MMS treatment.

From the results we showed in this MS, we conclude that the answer is yes: MMS phosphorylates Rad53 in telophase (Supplementary Fig. 3). This is clear when comparing to the telophase arrest. Nevertheless, the effect of MMS in telophase is much less pronounced than that observed in cells entering S-phase (compare Supplementary Fig. 3 and Fig 1a). In addition, phleomycin is much stronger than MMS in telophase (again compare Supplementary Fig. 3 and Fig 1a). Whether the absence of sister loci approximation in MMS is due to a sort of DDC finetuning or the fact of MMS is not expected to give rise to DSBs remains an interesting open question.

4. Line 275: I suggest to delete the word "all" since the authors show that Cdc14 is not responsible for Cin8 dephosphorylation.

Corrected.

5. Line 298: The word "must" is an overstatement. The role of Cin8 could also be passive. I therefore suggest to change "must" to "could".

Corrected.

6. Line 419: I disagree that the authors have shown that Cin8 relocalization is "sufficient" for the observed regression. The authors have not presented conditions where Cin8 relocalization takes place but none of the other events.

We have erased this sentence.

7. Line 429-430: What is the evidence for "further post-translational modifications" of Cin8?

This is part of the discussion. We just wanted to make clear that our results suggest that there are post-translational modifications (PTMs) that drive Cin8 relocalization, and that these PTMs include both the motor domain and residues out of the motor domain. It makes sense that some PTMs (probably already seen in the Western blots shown in Fig. 6b) correspond to those reported in the motor domain because the Cin8-3D version is partly impaired in relocalizing (Fig. 6c). However, since (i) Cin8-3D can still partly relocalize and (ii) Cin8-3A fully relocalizes, we believe that there must be PTMs out of the motor domain as well. Lastly, but not least, the fact that Cin8 still gets dephosphorylated in *cdc14-1* (Supplementary Fig. 9) is in agreement with PTMs out of the motor domain (at least the three CDK residues previously reported).

Reviewer #3 (Remarks to the Author):

This revised version is overall a much better and a much more convincing paper. My main concerns were appropriately addressed.

I only advise to place the important I-SceI result (currently figure S8) within one of the main figures.

Thanks for the advice. Indeed, we initially thought of placing this result within one main figure. We chose not because: (i) it should go together with all other results that deal with sister loci coalescence, and this figure (Fig. 3) is already saturated; (ii) although I-SceI strengthens the statement that DSBs are the cause of coalescence, we have used this system only to reinforce such statement, whereas the rest of the MS deals with random DSBs generated with phleomycin. Therefore, to us, it makes much more sense to focus on phleomycin throughout all main figures.